# Flexible use of memory by food-caching birds

Marissa C Applegate, Dmitriy Aronov*

Mortimer B. Zuckerman Mind Brain Behavior Institute, Columbia University, New York, United States

**Abstract** Animals use memory-guided and memory-independent strategies to make navigational decisions. Disentangling the contribution of these strategies to navigation is critical for understanding how memory influences behavioral output. To address this issue, we studied spatial behaviors of the chickadee, a food-caching bird. Chickadees hide food in concealed, scattered locations and retrieve their caches later in time. We designed an apparatus that allows birds to cache and retrieve food at many sites while navigating in a laboratory arena. This apparatus enabled automated tracking of behavioral variables – including caches, retrievals, and investigations of different sites. We built probabilistic models to fit these behavioral data using a combination of mnemonic and non-mnemonic factors. We found that chickadees use some navigational strategies that are independent of cache memories, including opportunistic foraging and spatial biases. They combine these strategies with spatially precise memories of which sites contain caches and which sites they have previously checked. A single memory of site contents is used in a context-dependent manner: during caching chickadees avoid sites that contain food, while during retrieval they instead preferentially access occupied sites. Our approach is a powerful way to investigate navigational decisions in a natural behavior, including flexible contributions of memory to these decisions.

## Editor's evaluation

The extreme memory capacities of food-caching birds provide untapped opportunities for studying mechanisms of memory formation and retrieval. Here, Applegate and Aronov develop an automated animal and cache-site tracking system in which moments of seed deposits, retrievals, and checks are measured continuously alongside the animal's spatial positions. Probabilistic models reveal idiosyncratic spatial preferences in individual birds and also identify flexible memory usage – in which a single memory of past seed deposition can differentially guide spatial trajectories depending on if the bird is in engaged in retrieving or storing seeds. The rigorous behavioral tracking and modeling sets the stage for dissection of neural mechanisms underlying memory storage and retrieval.

**\*For correspondence:** da2006@columbia.edu

**Competing interest:** The authors declare that no competing interests exist.

## Introduction

The study of episodic memory in animals has been greatly benefitted by experiments that involve spatial navigation (*DeVito and Eichenbaum, 2010*; *Morris et al., 1982*; *Salwiczek et al., 2010*). Whereas memory use is internal to the animal and not directly observable, navigational decisions that require memory are overt and trackable. In addition, memory-related neural circuits, like the hippocampus, exhibit well-described firing patterns that correlate to navigational variables – including place and head direction (*O'Keefe and Dostrovsky, 1971*; *Taube et al., 1990*). However, navigation is a rich behavior that includes both mnemonic and non-mnemonic spatial strategies. For example, an animal using memory to obtain a reward may also engage in opportunistic foraging, use memory-independent search strategies, and exhibit spatial biases (*Dally et al., 2006*; *Kamil and Roitblat,*

**eLife digest** Humans form new memories about what is happening in their lives every day. These autobiographical memories depend on a part of the brain called the hippocampus. But how these memories are recorded remains unclear. Studying certain birds may help to provide more insight.

Black-capped chickadees, for example, are memory specialists. They stash thousands of food items and use their memories to recover these hidden food stores. This behavior also relies on these birds' hippocampus. Studying these animals' behavior in the laboratory may help scientists decode how the birds use their memories and to gain more insight about the brain processes underlying memory.

Now, Applegate and Aronov show that chickadees use memory not only to retrieve food but also to decide where to hide it in the first place. In the experiments, chickadees were placed in a special- ized enclosure with a grid of holes covered by silicone rubber flaps on the floor. The birds lifted the flaps with their toes or beak to hide a piece of sunflower seed underneath. Applegate and Aronov recorded and analyzed the animals' seed hiding and retrieving behavior with a video camera to deter- mine whether the birds were remembering the sites or happening on them by chance.

This revealed that black-capped chickadees use the same memories of where they had hidden food in two different ways. When they were hiding new morsels, the birds remembered where they had stashed food and avoided those flaps. When they were retrieving food, the birds knew exactly which flaps to look under. Future experiments using this special enclosure may help scientists monitor what happens in the chickadees' brains during these activities.

*1985*; *Krebs et al., 1974*; *Lamprea et al., 2008*; *Pravosudov, 2001*). It is critical to tease apart these contributions to behavior in order to eventually understand the underlying neural mechanisms.

Food-caching birds offer an opportunity to study memory in the context of navigation. These birds hide food in many distinct, concealed locations throughout their environment and later retrieve their caches. Food caching has been studied in two general types of settings well-suited for characterizing different components of the behavior. In some experiments, spatial navigation is minimized, and birds choose from a small number of available options to obtain food (*Brodbeck et al., 1992*; *Clayton and Dickinson, 1998*; *Clayton and Krebs, 1994a*; *Salwiczek et al., 2010*). These tasks have been ideal for characterizing various contributions to memory, including location, content, and even the relative time of different caches. In contrast, other experiments have been performed in the wild or in large, natu- ralistic settings (*Cowie et al., 1981*; *Herz et al., 1994*; *Krushinskaya, 1966*; *Sherry, 1984a*; *Stevens and Krebs, 2008*). These studies have been well-suited for measuring spatial properties of the food- caching behavior, including the distribution of caches, large-scale biases, and the spatial specificity of cache memories. However, it is unknown how mnemonic and other spatial strategies are coordinated by the animal within a single behavior.

Addressing this question may be possible in reduced spatial settings, like those typical in rodent experiments (*Muller et al., 1987*; *Poucet et al., 2003*; *Tolman, 1948*). Such a setting must retain key spatial aspects of bird navigation, while also incorporating food-caching behavior and memory use. Ideally, it would also permit detailed tracking of the animal's behavior and be compatible with neural recordings. We developed an experimental setup to fulfill these requirements for a food- caching species, the black-capped chickadee. We then used this setup to dissect the contributions of mnemonic and non-mnemonic navigational strategies to food caching, using probabilistic modeling of behavioral choices (*Brunton et al., 2013*; *Raposo et al., 2012*; *Scott et al., 2017*).

## Results

### Design of the behavioral paradigm

Our first goal was to engineer a behavioral setup in which chickadees navigate, cache food, and retrieve caches. We designed individual cache sites as holes in the floor of an arena covered by sili- cone rubber flaps (*Figure 1*). Dimensions and materials were chosen to allow chickadees to pull the flap open with little effort – using either their toes or their beak – and to deposit a piece of a sunflower seed underneath. Once released, the flap obstructed any visual access to the contents of the site from above. Flaps ensured that no visually guided, memory-independent strategy could be used by the

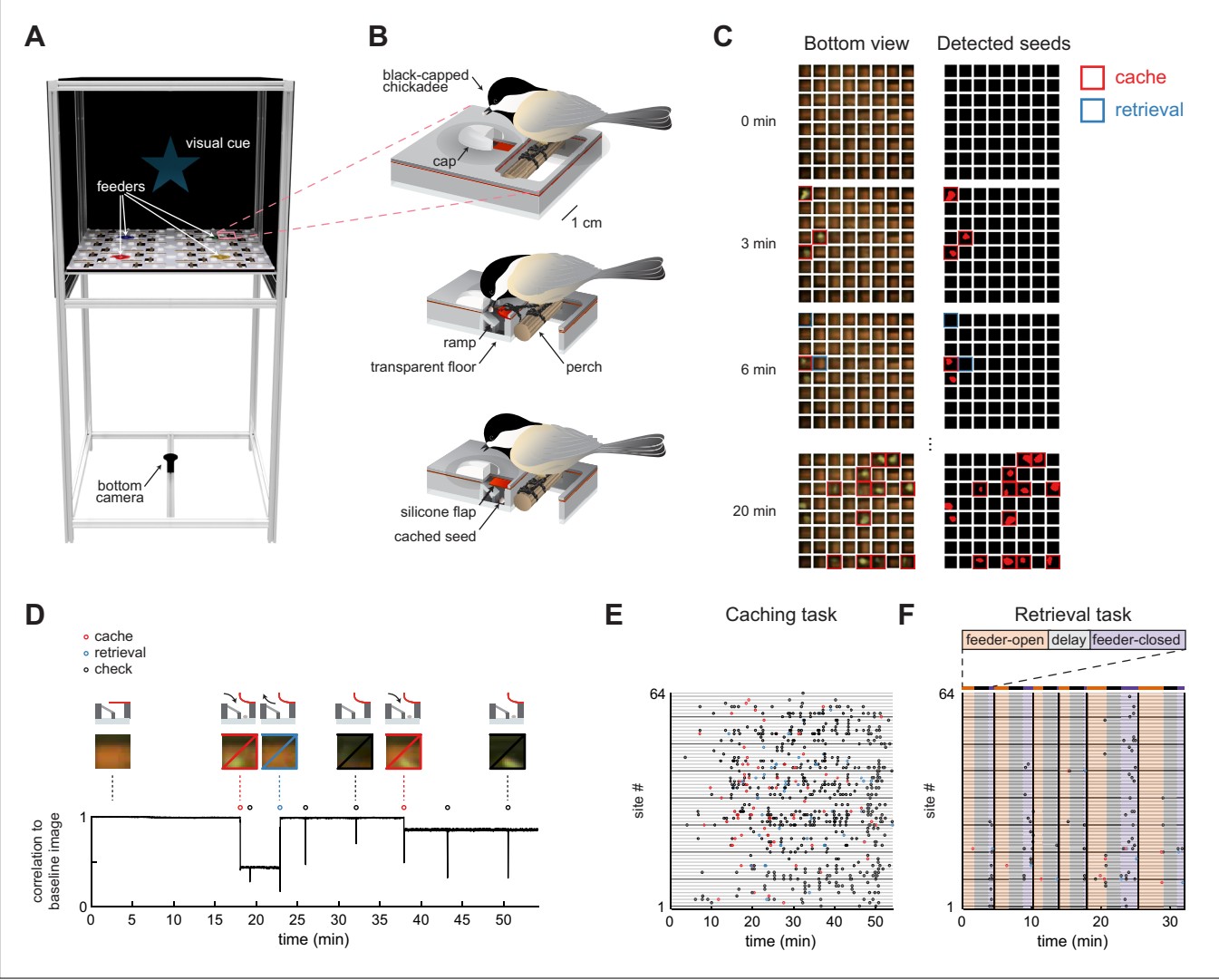

**Figure 1.** Behavioral paradigm for food caching and retrieval in chickadees. (**A**) Rendering of the behavioral setup. For clarity, the front wall that contains doors of the arena is removed. Pink box highlights one cache site.(**B**) Illustration of a chickadee at a cache site. Top: chickadee prior to caching. Middle: cross-section of the cache site, with a chickadee pulling open the silicone flap to deposit a seed. Bottom: cross-section of the cache site after the seed was cached. (**C**) Left: video frames from the bottom camera showing all 64 cache sites at four timepoints within a behavioral session. Right: real-time detection of cached seeds. Red area indicates shape enclosed by the detected seed contour. (**D**) Detection of events at one example cache site in one behavioral session. Top: cartoon of the bird's interactions with the site at several time points. Middle: video frames from the bottom camera at the corresponding time points. Bottom: Pearson correlation of each video frame with the image of the same cache site when empty. Caches create sustained decreases in the correlation, whereas site checks create transient decreases. (**E**) Ethogram of all behavioral events in an example session of the Caching task. Colored circles correspond to caches, retrievals, and site checks, as in (**D**). (**F**) Same as (**E**), for the Retrieval task. Colored regions indicate the phase of the trial. Black vertical lines denote trial boundaries.

The online version of this article includes the following video for figure 1:

**Figure 1—video 1.** Caching task.

https://elifesciences.org/articles/70600/figures#fig1video1

**Figure 1—video 2.** Retrieval task.

https://elifesciences.org/articles/70600/figures#fig1video2

chickadee to determine site contents. Sixty-four cache sites were arranged in an 8 × 8 grid inside a 61 × 61 cm square arena. In addition, the arena contained a food source in each of the quadrants: a feeder with a motorized cover that could be individually opened or closed. This setup allowed navigation on a similar spatial scale to that typically studied in rodents (*Poucet et al., 2003*) while also offering birds a large number of concealed cache sites.

We next developed methods to automatically track chickadee behavior in the caching arena. Automated behavioral tracking was necessary to allow closed-loop manipulations for some of the experiments described below. To achieve this, we used a transparent material for the bottom layer of the arena and positioned a video camera underneath. We applied a real-time contour detection algorithm (*Lopes et al., 2015*, see Materials and methods) to determine whether each of the sites was empty or occupied by a seed (*Figure 1C*). During offline analysis, we also detected instances of the bird opening the cover flap of a site, which we call 'checks'. To do this, we calculated the Pearson correlation between the image of each site when empty and every video frame of that site in the session. Periods of time when a site was occupied corresponded to sustained periods of low correlation, while checks corresponded to transient decreases in correlation (*Figure 1D*). Thus, caches, retrievals, and checks could all be detected in the behavior using a single-camera video recording. We also trained a deep neural network (*Mathis et al., 2018*; *Nath et al., 2019*) to track the bird's location for this analysis.

We designed two tasks, each suited for analyzing different aspects of behavior (for details see Materials and methods). In the *Caching task* (*Figure 1E*, *Figure 1—video 1*), chickadees were free to cache without any imposed trial structure for 1 hr. Individual feeders opened and closed to motivate movement through the arena and caching, but the schedule of these openings and closings was unrelated to the bird's behavior. Our term 'Caching task' does not mean that birds only cached in this task – in fact, birds also ate seeds, checked sites, and retrieved their caches. However, because this task placed no limit on the number of caches, it was particularly well-suited for investigating the chickadee's choice of where to cache each seed. This task was performed by 17 chickadees. In 10 of these, we recorded enough caches (> 64; see Materials and methods) for inclusion in the remainder of the Results section.

In the *Retrieval task* (*Figure 1F*, *Figure 1—video 2*), a trial structure was imposed. At the beginning of each trial, one of the feeders was open ('feeder-open phase'), allowing chickadees to eat or cache seeds. Once the bird cached 1–3 seeds, lights were turned off for a 2 min delay phase. After the delay, all feeders were closed ('feeder-closed phase'), and the only sources of food in the arena were previously cached seeds. In the feeder-closed phase, birds were generally motivated to find and retrieve their caches. This task was therefore suitable for investigating which sites the chickadee choses to check during cache retrieval. This task was performed by seven chickadees, all of which are included in the analyses.

## Chickadees have unique and stable spatial biases

We first asked whether chickadees cached into some sites of the arena more than into others. In the Caching task, we calculated the cache distribution ($p^{\text{bias}}$) by dividing the number of caches into each of the 64 sites across sessions by the total number of caches (*Figure 2A*). Birds used most of the arena for caching (between 41–64 sites for each of the 10 birds), but in many cases cache distributions appeared non-uniform. To quantify this, we measured the entropy of $p^{\text{bias}}$ for each bird and compared its value to simulations in which the same number of caches were sampled from a uniform distribution (*Figure 2B*). In all birds, entropy was lower than expected by chance (p < 0.001). Therefore, birds exhibited spatial biases, even though they cached widely throughout the arena.

There are several possible explanations for the observed biases. One possibility is that biases are broadly shared across individuals of the species – akin to the preference for locations near walls in rodents (*Lamprea et al., 2008*). Another possibility is that biases are shared across some subgroups of individuals, but not all members of the species. Finally, biases could be unique to individual birds. To distinguish between these possibilities, we first asked whether there were common types of cache distributions across birds. We calculated principal components of all $p^{\text{bias}}$ values (*Figure 2C*). Remarkably, birds formed two clusters that were well-separated along the first principal component (*Figure 2—figure supplement 1A*). The first principal component resembled a bump in the center of the arena (*Figure 2C*), and accounted for 59% of the variance across individuals (*Figure 2—figure supplement 1B*). Thus, there were two groups of birds: those that preferred to cache in the center of the arena, and those that preferred the edges.

We next asked whether membership in one of these two groups was sufficient to explain the observed spatial biases. For each bird, we computed cache distributions separately for two randomly chosen, equal-sized subsets containing the same number of behavioral sessions. We then measured

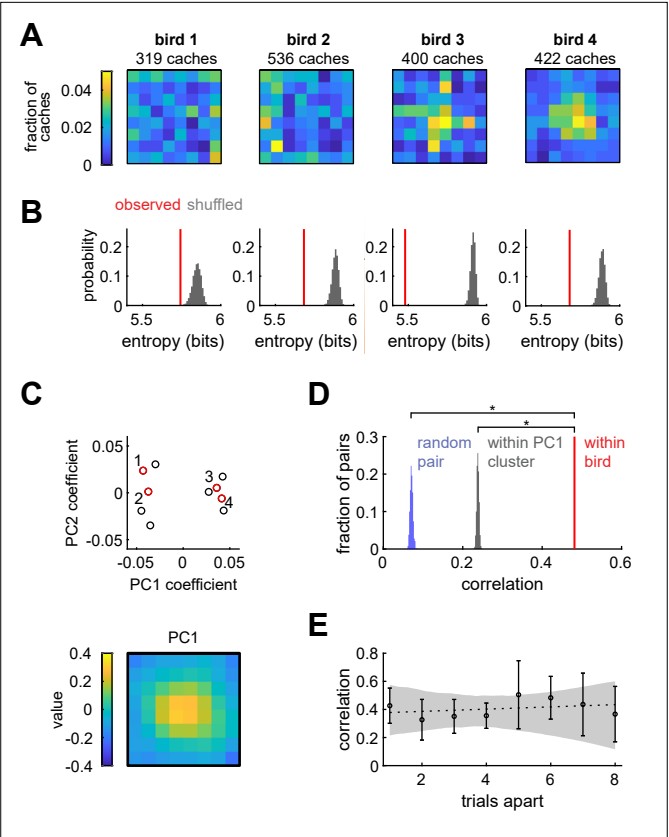

**Figure 2.** Chickadees exhibit idiosyncratic and stable biases in the locations of caches. (**A**) Probability distributions of cache locations across all sessions for four example birds. Distributions are denoted by $p^{bias}$ in the text. (**B**) Red lines: entropy values of the spatial distributions shown in (**A**). Gray histograms: entropy values for simulated caches drawn from a uniform distribution. Number of simulated caches was the same as in the observed data. (**C**) Principal component analysis of $p^{bias}$ values. Top: coefficients of the first two principal components for all birds. Red circles and numbers indicate birds shown in (**A**) and (**B**). Bottom: the first principal component. Birds cluster into center-preferring and edge-preferring groups. (**D**) Pearson correlation of $p^{bias}$ between subsets of all sessions paired within bird, between different birds, and between birds selected from the same PC1 cluster shown in (**C**). (**E**) Pearson correlation of $p^{bias}$ between pairs of trials at different trial lags. Values are medians across birds. Error bars: sem. Dashed line: linear regression. Gray shade: 95% confidence interval of linear regression fit to bootstrapped birds.

The online version of this article includes the following figure supplement(s) for figure 2:

**Figure supplement 1.** Validation of the clusters of $p^{bias}$ values (**A**) Assignment of birds to two clusters using k-means clustering of $p^{bias}$ values and its cross validation.

**Figure supplement 2.** Chickadees exhibit idiosyncratic and stable biases in the locations of checks.

the Pearson correlation between the two distributions from the same bird ('within-bird correlation') and between distributions from different, randomly paired birds ('across-bird correlation'). Within-bird correlation values were significantly higher than across-bird correlation values (p < 0.001, *Figure 2D*), even when birds were paired exclusively with birds from the same group (center-preferring or edge-preferring). Therefore, birds exhibited individual-specific biases that could not be entirely explained by common biases across a group of birds. An example of this is evident in *Figure 2A*: although birds 3 and 4 were both center-preferring birds, bird 3 had some additional bias for centrally-positioned rows and columns of the arena.

Finally, we asked whether spatial biases in individual birds were stable or drifted over time. We measured the Pearson correlation between cache distributions computed on pairs of sessions separated by different lags (*Figure 2E*). The slope of the relationship between lag and correlation was not significantly different from zero (p = 0.49, Bootstrap test), indicating that there was no detectable drift in bias.

We repeated all of the above analyses for the distribution of site checks in the Retrieval task. In this case, $p^{bias}$ was computed by dividing the number of times a particular site was checked by the total number of all site checks. All the results were similar to the ones obtained for caches in the Caching task (*Figure 2—figure supplement 2*). Collectively, our results suggest that chickadees exhibit idiosyncratic but stable biases, both in the sites that they check and those that they choose for caching. In both cases, the $p^{bias}$ distribution therefore serves as a valid baseline for subsequent models, described below.

## Probabilistic models of chickadee behaviors

How do chickadees choose where to cache? One model consistent with our analysis so far is that birds choose each cache location by drawing it randomly from the distribution $p^{bias}$. However, this is not the only possibility because $p^{bias}$ is computed by pooling across all time points. On a moment-by-moment basis, the likelihood of caching into a particular site may differ from the value given by $p^{bias}$. For example, the choice of cache site could depend on the proximity of a site to the bird: chickadees may tend to cache close to the location they last interacted with or, conversely, they may avoid the vicinity of that location. The choice of a cache site may also depend on which sites are already occupied by seeds, since birds may tend to either cluster or to spread out their caches. The bird's knowledge of which sites are occupied might be updated either by the act of caching, or by the act of checking the site to view its contents.

To model these possibilities, we considered three types of 'special' places in the arena at each moment in the Caching task: (1) the last site or feeder that the bird interacted with ('previous site'), which includes checking a site or retrieving a seed, (2) sites that are currently occupied by cached

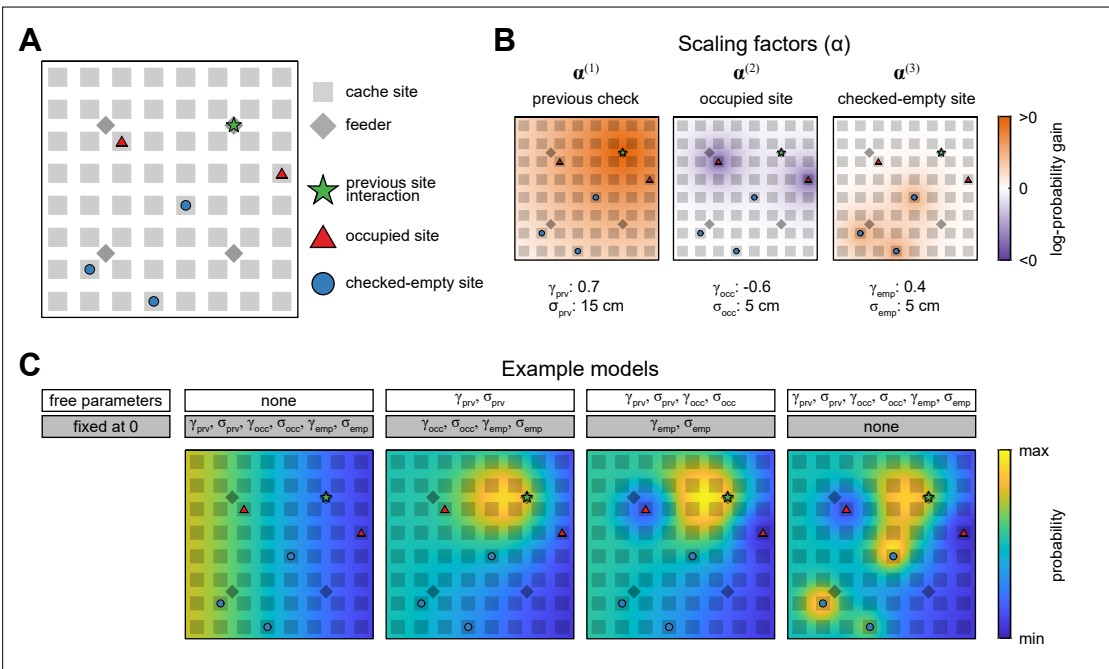

**Figure 3.** Schematic of the probabilistic behavioral models. Models here compute the probability of caching into a particular site in the Caching task. Equivalent models are used for the probability of checking a particular site in the Retrieval task. (**A**) Schematic of the arena at an example time point in the Caching task depicting 'special locations' in the arena: previous site interaction, occupied sites, and checked-empty sites. (**B**) Examples of scaling factors that change with proximity to each of the special locations shown in (**A**). Probability distribution of caching into different sites is computed by multiplying all scaling factors by $p^{bias}$. (**C**) Example models. For each model, some parameters are fixed at 0, while others are free and fit using maximum-likelihood estimation. For free parameters, examples shown here use the same values as in (**B**). All four models are plotted on the same color scale.

The online version of this article includes the following figure supplement(s) for figure 3:

**Figure supplement 1.** Non-linear interaction of terms.

**Figure supplement 2.** Using alternate shapes for the previous site scaling factor.

**Figure supplement 3.** Using different sigma values across the arena.

seeds ('occupied sites'), and (3) sites that have been previously checked by the bird and are currently empty ('checked-empty sites') (*Figure 3A*). We then constructed three scaling factors whose values changed with proximity to each of these special places (*Figure 3B*, defined in Materials and methods). The likelihood of caching into a particular site was computed by multiplying all three scaling factors by the corresponding value of $p^{\text{bias}}$ and normalizing the product to a sum of 1 across the arena. In other words, our model took each bird's individual biases into account and then asked whether caching probabilities were additionally influenced, on a moment-by-moment basis, by special places in the arena.

Each scaling factor was a Gaussian function across space, defined by two parameters. The first parameter was the amplitude of the Gaussian ($\gamma_{\text{prv}}$, $\gamma_{\text{occ}}$, or $\gamma_{\text{emp}}$ for the previous, occupied, and checked-empty sites, respectively), and indicated the strength and direction of the corresponding effect on probability, measured on a logarithmic base-10 scale. For example, $\gamma_{\text{occ}} = -1$ meant that the act of caching into a site decreased the likelihood of caching there again by a factor of 10. The second parameter was the width of the Gaussian ($\sigma_{\text{prv}}$, $\sigma_{\text{occ}}$, or $\sigma_{\text{emp}}$, respectively) and indicated the spatial extent of each effect. In other words, a large value of $\sigma_{\text{occ}}$ meant that the act of caching into a site affected the likelihood of subsequently caching into neighboring sites as well. This model was first applied to the data obtained in the Caching task. For each bird, we optimized the values of the model parameters in order to maximize the likelihood of the experimentally observed caches (*Brunton et al., 2013*; *Raposo et al., 2012*; *Scott et al., 2017*).

We also asked how chickadees chose which sites to check in the Retrieval task. Much like the choice of cache location, the choice of check location may be affected by proximity to the previous site, to occupied sites, and to checked-empty sites. We therefore used the same model as above, but fit to locations of checks instead of caches. At first, we fit the model only to the checks that the bird made up until and including finding a cache in the Retrieval task. We did this because after finding the first cache, birds often recached food, which confounded the analysis by mixing caching and retrieval behaviors in the same time period. We will analyze recaching separately later.

Our approach was to start with a baseline model (Model #0), in which the likelihood of choosing a particular site for caching or checking was given by $p^{\text{bias}}$ (*Figure 3C*). In this model, there were no free parameters, and the values of $\gamma_{\text{prv}}$, $\gamma_{\text{occ}}$, $\gamma_{\text{emp}}$, $\sigma_{\text{prv}}$, $\sigma_{\text{occ}}$, and $\sigma_{\text{emp}}$ were all set to 0. We then gradually introduced free parameters to the model, one or two at a time. With each parameter introduction, we tested whether model performance was improved by measuring the Akaike Information Criterion (AIC), which quantifies model likelihood accounting for the number of free parameters (*Dziak et al., 2020*; *Lubke et al., 2017*).

Our model makes several assumptions. First, it uses a Gaussian function for each of the scaling factors. Second, it assumes a uniform spatial extent of this Gaussian across the arena. Finally, by multiplying scaling factors together it assumes that they have independent effects on probability. In separate models, we tested these assumptions and showed that chickadee behavior can be explained even better by including nonlinear interactions between scaling factors (*Figure 3—figure supplement 1*) by using functions that have more complex shapes than a Gaussian (*Figure 3—figure supplement 2*), and by allowing different spatial extents in different parts of the arena (*Figure 3—figure supplement 3*). However, these improvements were relatively modest and did not affect the main results described below.

## Chickadee behaviors exhibit a strong proximity effect

We first asked how the location of the previous site affected the chickadee's behavior. For this analysis we constructed Model #1, in which $\gamma_{\text{prv}}$ and $\sigma_{\text{prv}}$ were introduced as free parameters (*Figure 4*). We found that this model performed significantly better than Model #0 (see *Table 1* for statistics of all model comparisons). The best-fit value of $\gamma_{\text{prv}}$ was positive in all birds (0.94 ± 0.21), and the best-fit value of $\sigma_{\text{prv}}$ was 15.5 ± 0.9 cm (see *Table 2* for statistics of all model parameters). Thus, there was roughly a 10-fold (i.e., $10^{0.94}$) increase in probability of caching next to the previous site, and the spatial extent of this effect was about 1/4 of the arena. To verify this result independently of the model, we compared the observed and expected distances of caches to previous sites and indeed found that birds cached closer to previous sites than expected by chance given each bird's spatial bias (*Figure 4—figure supplement 1*).

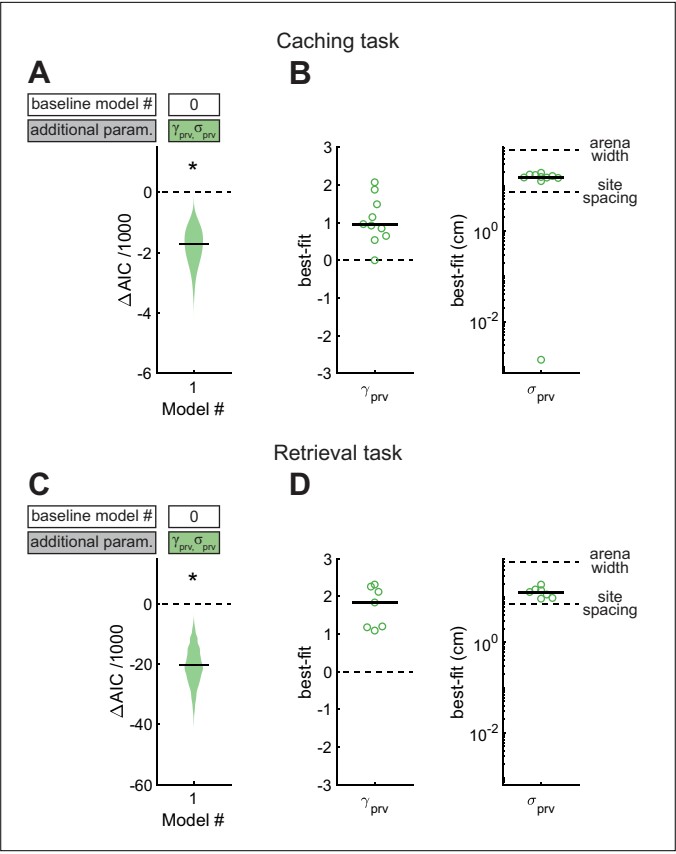

**Figure 4.** Effect of proximity to the previous site in Caching and Retrieval tasks. (**A**) Comparison of Model #1 to Model #0 applied to the Caching task. Here and in subsequent figures, ΔAIC indicates the difference in AIC between the model being considered (in this case Model #1, labeled on the x-axis) and the baseline model (in this case Model #0, labeled in the white box). Compared to Model #0, Model #1 uses two additional free parameters ($\gamma_{prv}$ and $\sigma_{prv}$), specified in the colored box. Horizontal black line: ΔAIC value for caches pooled from all birds. Shaded area: distribution of 1000 ΔAIC values on data bootstrapped across birds. Values less than 0 indicate model improvement. Asterisk indicates statistically significant improvement. (**B**) Best-fit values of model parameters prv and σprv for Model #1 applied to the Caching task. Symbols indicate values for individual birds. Black line: median values across birds. (**C, D**) Same as (**A, B**), but for models applied to the Retrieval task.

The online version of this article includes the following figure supplement(s) for figure 4:

**Figure supplement 1.** Corroboration of the proximity effect.

We applied the same model to site checks in the Retrieval task. Again, Model #1 performed better than Model #0. The value of $\gamma_{prv}$ was positive in all birds (1.83 ± 0.42), indicating a tendency to check sites close to the previous sites, and $\sigma_{prv}$ was 12.9 ± 1.7 cm. This result is consistent with our qualitative observations of chickadees in the Retrieval task: as birds moved through the arena – sometimes in the direction of a hidden cache – they often checked sites on their path. In fact, the value of $\gamma_{prv}$ was significantly higher in the Retrieval task (p < 0.05, unpaired t-test), but the best-fit $\sigma_{prv}$ was not significantly different between the two tasks (p = 0.54, unpaired t-test). These results show that birds are influenced by proximity during retrieval more than during caching, even though the spatial extent of this effect is similar during the two behaviors.

## Chickadee behaviors are affected by site content

We next asked whether chickadee behaviors were additionally influenced by the contents of individual sites (*Figure 5A–B*). In the Caching task, we fit Model #2, which included $\gamma_{occ}$ as a third free parameter to model the effect of occupied sites. Model #2 performed even better than Model #1, indicating that occupied sites indeed affected the behavior. We then fit Model #3, in which $\gamma_{emp}$ was additionally introduced as a fourth free parameter to model the effect of checked-empty sites. Model

#3 performed better yet, indicating that checked-empty sites also affected the behavior. Across birds, the value of $\gamma_{occ}$ was negative (–0.32 ± 0.17), whereas the value of $\gamma_{emp}$ was positive (0.13 ± 0.05). These values indicate roughly a twofold decrease in probability of caching into occupied sites and a 1.3-fold increase in probability of caching into checked-empty sites. Both results are consistent with chickadees spreading their caches throughout the arena, rather than pooling them into the same

**Table 1.** Summary of model performance.

Statistical analysis for all 12 models (0 through 11) presented in the Results section. Each model is compared to one of the previous models using bootstrap analysis of AIC values (see Materials and methods). p-values < 0.05 are emphasized in bold. Model #0 does not have corresponding p-values because it was the baseline model that was not compared to anything. Some models were only fit to data from the Caching or Retrieval task; in these cases, a hypen indicates that the corresponding fit was not performed.

| Model # | Free parameters | Compared to Model # | Model improvement? Caching task p-value | Model improvement? Retrieval task p-value |
|---|---|---|---|---|
| 0 | none | - | - | - |
| 1 | $\gamma_{prv}$; $\sigma_{prv}$ | 0 | **< 0.001** | **< 0.001** |
| 2 | $\gamma_{prv}$; $\sigma_{prv}$; $\gamma_{occ}$ | 1 | **< 0.001** | **< 0.001** |
| 3 | $\gamma_{prv}$; $\sigma_{prv}$; $\gamma_{occ}$; $\gamma_{emp}$ | 2 | **< 0.001** | **0.015** |
| 4 | $\gamma_{prv}$; $\sigma_{prv}$; $\gamma_{occ}$; $\gamma_{emp}$; $\sigma_{occ}$ | 3 | 0.48 | 0.35 |
| 5 | $\gamma_{prv}$; $\sigma_{prv}$; $\gamma_{occ}$; $\gamma_{emp}$; $\sigma_{emp}$ | 3 | 0.93 | 0.42 |
| 6 | $\gamma_{prv}$; $\sigma_{prv}$; $\gamma_{occ}^{c}$ | 1 | - | **< 0.001** |
| 7 | $\gamma_{prv}$; $\sigma_{prv}$; $\gamma_{occ}^{c}$; $\gamma_{occ}^{r}$ | 6 | - | **< 0.001** |
| 8 | $\gamma_{prv}$; $\sigma_{prv}$; $\gamma_{occ}$; $\gamma_{emp}$; $\tau_{occ}$ | 3 | 0.48 | - |
| 9 | $\gamma_{prv}$; $\sigma_{prv}$; $\gamma_{occ}$; $\gamma_{emp}$; $\tau_{emp}$ | 3 | 0.24 | - |
| 10 | $\gamma_{prv}$; $\sigma_{prv}$; $\gamma_{occ}$; $\gamma_{emp}$; $\nu_{occ}$ | 3 | 0.97 | - |
| 11 | $\gamma_{prv}$; $\sigma_{prv}$; $\gamma_{occ}$; $\gamma_{emp}$; $\nu_{emp}$ | 3 | 0.29 | - |

**Table 2.** Contributions of different factors to behavior.

Statistical analysis of all $\gamma$ parameters mentioned in the Results section, which indicate the contribution of different factors to behavior. Best fit parameter values are indicated as median ± standard error of the median across all birds (N = 10 in the Caching task and N = 7 in the Retrieval task). Significance across birds is calculated using a one-sided Wilcoxon rank sum test to determine whether values across the population are significantly different from 0. The fraction of birds that are individually significant is calculated by bootstrapping the behavioral sessions of each bird. A p-value of 0.05 is used to assign significance in this case. See Materials and methods for details.

| Parameter | Best-fit parameter value Caching task | Best-fit parameter value Retrieval task | Significance across birds Caching task p-value | Significance across birds Retrieval task p-value | Fraction of individually significant birds Caching task | Fraction of individually significant birds Retrieval task |
|---|---|---|---|---|---|---|
| $\gamma_{prv}$ | 0.94 ± 0.21 | 1.83 ± 0.42 | **< 0.001** | **< 0.001** | 8/10 | 7/7 |
| $\gamma_{occ}$ | –0.32 ± 0.17 | 0.26 ± 0.09 | **< 0.001** | **0.008** | 8/10 | 7/7 |
| $\gamma_{emp}$ | 0.13 ± 0.05 | –0.14 ± 0.07 | **0.005** | **0.008** | 6/10 | 5/7 |
| $\gamma_{occ}$, first check | –0.28 ± 0.17 | 0.18 ± 0.10 | **0.005** | **0.008** | 8/10 | 2/7 |
| $\gamma_{emp}$, first check | 0.12 ± 0.05 | - | **0.042** | - | 6/10 | - |
| $\gamma_{occ}^{c}$ | - | 0.42 ± 0.05 | - | **0.008** | - | 7/7 |
| $\gamma_{occ}^{r}$ | - | 0.26 ± 0.07 | - | **0.008** | - | 6/7 |

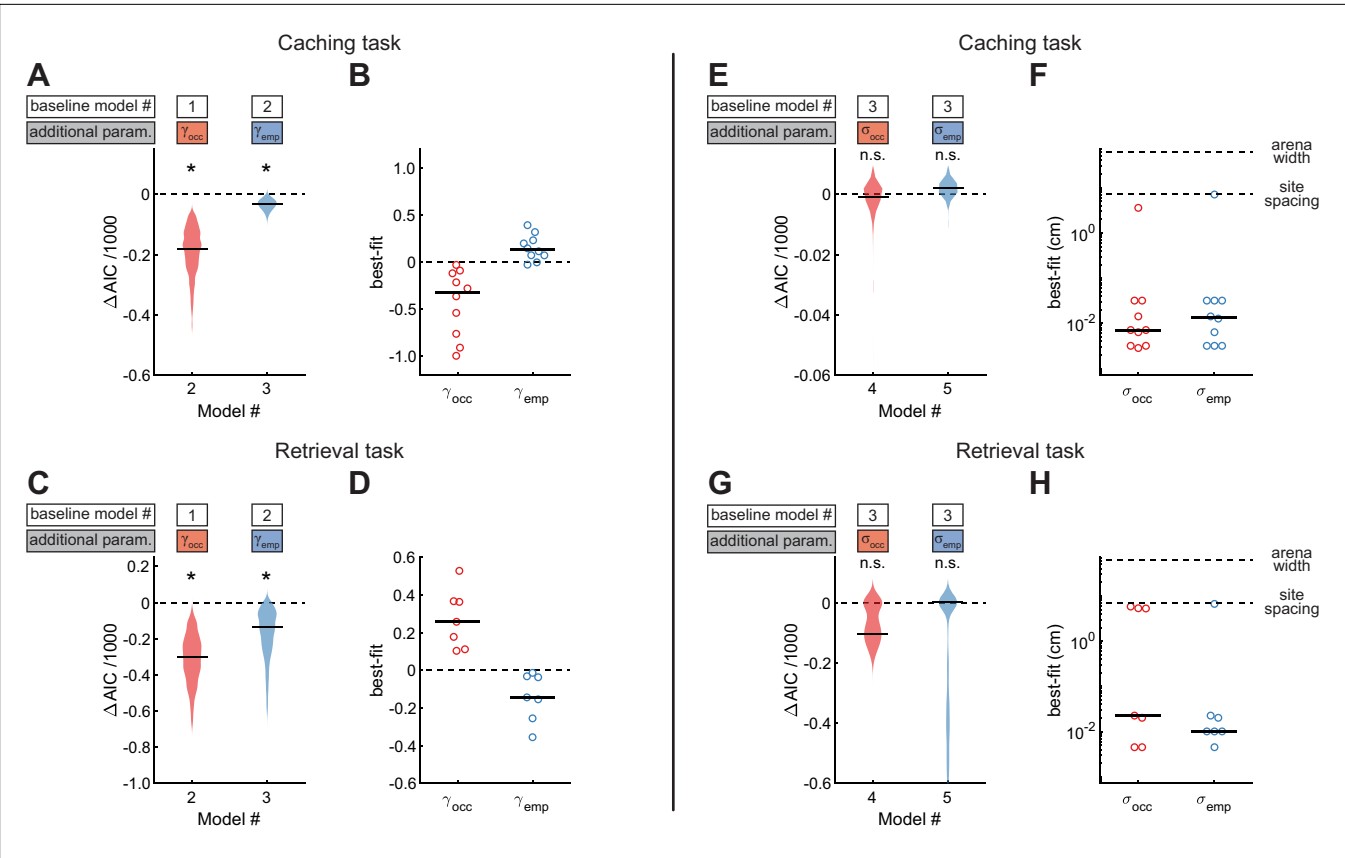

**Figure 5.** Site-specific, opposing effects of site content on behavior in Caching and Retrieving tasks. (**A**) Performance of Models #2 and #3 applied to the Caching task, plotted as in *Figure 4A*. These models introduce free parameters to quantify the effect of occupied and checked-empty sites on behavior. (**B**) Best-fit values of $\gamma_{occ}$ and $\gamma_{emp}$ applied to the Caching task, plotted as in *Figure 4B*. (**C, D**) Same as (**A, B**) but for models applied to the Retrieval task. (**E**) Performance of Models #4 and #5 applied to the Caching task, plotted as in *Figure 4A*. These models introduce free parameters to quantify the spatial extent of the effect of occupied and checked-empty sites on behavior. (**F**) Best-fit values of $\sigma_{occ}$ and $\sigma_{emp}$ applied to individual birds in the Caching task, plotted as in *Figure 4B*. (**G, H**) Same as (**E, F**), but for models applied to the Retrieval task.

The online version of this article includes the following figure supplement(s) for figure 5:

**Figure supplement 1.** Corroboration of the effects of specific site contents on behavior.

sites. Indeed, chickadees tended to cache into occupied sites less often than expected by chance (*Figure 5—figure supplement 1A*).

We again applied the same models to site checks in the Retrieval task (*Figure 5C–D*). As in the Caching task, introducing $\gamma_{occ}$ and $\gamma_{emp}$ significantly improved the fit of the model. However, the signs of the two parameters were reversed compared to the Caching task: $\gamma_{occ}$ was positive in all birds (0.26 ± 0.09), whereas $\gamma_{emp}$ was negative in all birds (–0.14 ± 0.07). These results show that birds were attracted to occupied sites during the feeder-closed phase of the task. This is consistent with chickadees searching for their caches. Indeed, chickadees found their caches after fewer checks than expected by chance: 8.5 ± 5.6 checks in the observed data, compared to 20.8 ± 4.5 checks in data where trajectories were shuffled between trials (*Figure 5—figure supplement 1B*).

Our analyses so far show that chickadees can be attracted to or avoid specific sites depending on the content of these sites. We asked whether these behavioral effects were site-specific, or whether neighboring locations were affected as well. We introduced either $\sigma_{occ}$ or $\sigma_{emp}$ as an additional free parameter (Model #4 and Model #5 respectively, *Figure 5E–H*). Neither model was an improvement over Model #3. In fact, the best-fit values of $\sigma_{occ}$ and $\sigma_{emp}$ in both models were between 0.01 and 0.02 cm, much smaller than the distance between neighboring sites (7.1 cm). Thus, the effects of site contents on chickadee behavior were site-specific, and were small even at neighboring sites of the arena. We verified this result independently of the model by measuring the fraction of caches and checks at various distances away from occupied sites (*Figure 5—figure supplement 1C–D*). This site

specificity is consistent with that demonstrated under different conditions by previous work (*Cowie et al., 1981*).

## Chickadees have memories of site contents

We have shown that chickadees avoid caching into sites that are occupied, but increase their probability of caching into checked-empty sites. An intriguing explanation of these results is that chickadees use memory of site contents to influence their caching behavior. However, after obtaining a seed, a chickadee may simply check multiple sites in the arena and preferentially cache into an empty one that it encounters. This would be a visually-guided strategy that does not require memory. We considered this possibility unlikely because chickadees usually cached into the first site that they opened after visiting the food source (79% of caches). However, we wanted to ensure that the remaining 21% of caches were not responsible for the observed effects. We implemented the same models as before (Models #2 and #3), but instead of fitting them to the observed caches, we fit them to the locations of the first sites opened by the bird after obtaining a seed ('first check'). We found that $\gamma_{occ}$ in Model #2 was still significantly negative, and $\gamma_{emp}$ in Model #3 was still significantly positive (*Figure 6A–B*). Therefore, chickadees did not merely check sites until finding an empty one. Rather, they actually used memories of which sites were occupied and which were empty.

We also fit Model #2 to first checks in the Retrieval task. In this case, $\gamma_{occ}$ was still positive. In other words, chickadees had an increased probability of checking an occupied site even on the first attempt. This implies that chickadees did not search the arena until finding a cache, but used memories of occupied sites. Note that $\gamma_{emp}$ was not computed for this analysis because, by definition, there could not be any checked-empty sites before the first check.

Finally, we wanted to make sure that chickadees did not use olfactory cues to detect seeds in cache sites. Chickadees are not known to use olfaction for finding food (*Cowie et al., 1981*; *Herz et al., 1994*; *Sherry, 1984a*), and previous studies have excluded this possibility by removing caches during the delay period of the task. We repeated this experiment in our arena. Consistent with published results, even when caches were removed from the arena, chickadees tended to spend more time at the locations of the missing caches after a delay period of up to 1 hr (*Figure 6—figure supplement 1*). Collectively, our results show that chickadees have memories of site contents, as in previous work (*Clayton and Dickinson, 1999*; *Sherry, 1984a*). They use these memory of site contents in flexible ways, both during caching and during retrieval.

## Memories of site contents last the duration of the session

We next asked for how long memories lasted in our arena. In the feeder-closed phase of the Retrieval task, chickadees often 'recached' seeds after retrieving them (44% of all retrievals). Therefore, there were two types of caches in this task: recent recaches and older caches made prior to the delay period. We first analyzed how these different types of caches affected behavior. For this analysis, we introduced models with separate parameters $\gamma_{occ}^{c}$ and $\gamma_{occ}^{r}$ instead of $\gamma_{occ}$ to fit the contribution of older caches and recaches, respectively. These models were fit to the entire feeder-closed phase of the Retrieval task, which contained both the retrieval and the recaching behaviors. Model #6 introduced $\gamma_{occ}^{c}$ as a free parameter, while Model #7 introduced both $\gamma_{occ}^{c}$ and $\gamma_{occ}^{r}$. We found that Model #6 fit the data significantly better than Model #1, and Model #7 fit the data even better than Model #6 (*Figure 6D*). The best-fit values were significantly positive for both parameters, but higher for $\gamma_{occ}^{r}$ than for $\gamma_{occ}^{c}$ (0.42 ± 0.05 vs 0.26 ± 0.07, respectively, $p < 0.01$ for the comparison). These results indicate that birds were attracted to all of their caches during retrieval, but had a stronger preference for the recent recaches.

One possible explanation of this result is that birds had a memory decay that reduced the effect of older caches on behavior. However, birds often recache a single seed multiple times, and may simply have a preference for revisiting recache locations. We therefore performed a different analysis that modeled a continuous memory decay. The Caching task was ideally suited for this analysis because it did not have a trial structure; caches could be retrieved at any later time in the session. We defined a 'memory decay factor' for each site, which was reset to one whenever that site was checked and decayed exponentially to 0 afterwards. In Model #8, this factor was multiplied by $\gamma_{occ}$ and decayed with a timescale $\tau_{occ}$. In Model #9, this factor was multiplied by $\gamma_{emp}$ and decayed with a timescale $\tau_{emp}$. Both models were compared to Model #3, in which neither $\gamma_{occ}$ nor $\gamma_{emp}$ decayed. We found that

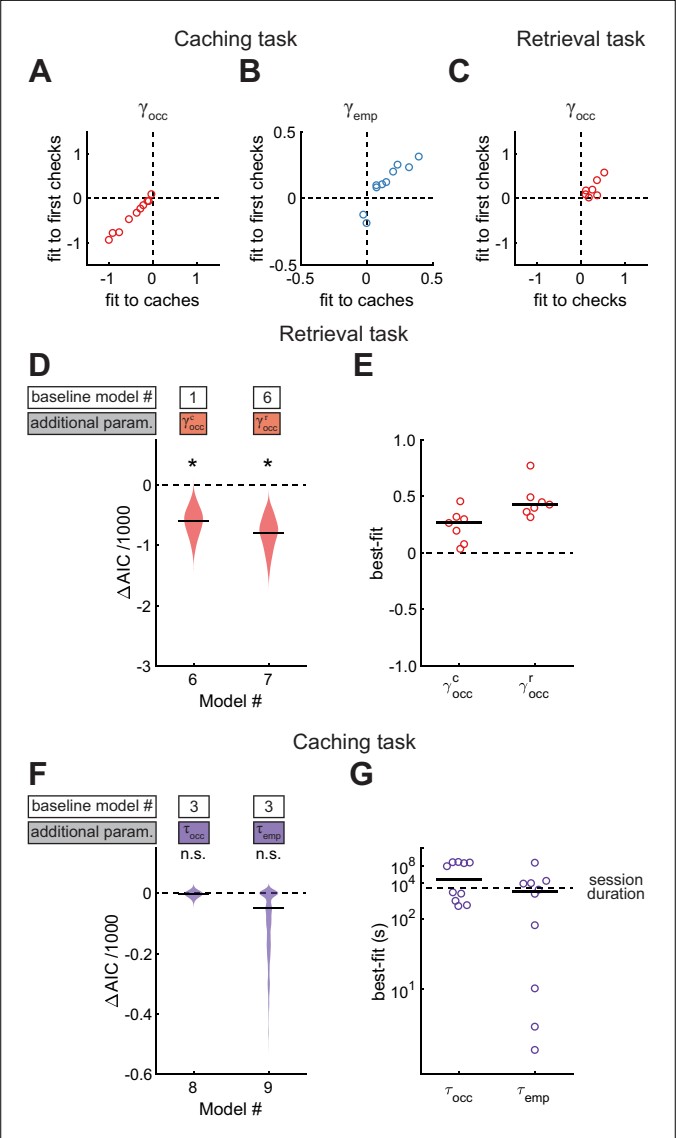

**Figure 6.** Results are best explained by long-lasting memories. (**A, B**) Values of parameters across birds, compared between models applied to caches or first checks in the Caching task. Values of $\gamma_{occ}$ are from Model #2, and values of $\gamma_{emp}$ are from Model #3. (**C**) Values of $\gamma_{occ}$ across birds, compared between Model #2 applied to checks or the same model applied to first checks in the Retrieval task. (**D**) Performance of Models #6 and #7 applied to the Retrieval task, plotted as in *Figure 4A*. These models introduce free parameters to separately quantify the effects of caches and recaches on the site checking behavior. (**E**) Best-fit values of $\gamma_{occ}^{c}$ and $\gamma_{occ}^{r}$ applied to individual birds in the Retrieval task, plotted as in *Figure 4B*. (**F**) Performance of Models #8 and #9 applied to the Caching task, plotted as in *Figure 4A*. These models introduce free parameters $\tau_{occ}$ and $\tau_{emp}$ to quantify the decay of $\gamma_{occ}$ and $\gamma_{emp}$ over time since the site was last checked by the bird. (**G**) Best-fit values of $\tau_{occ}$ and $\tau_{emp}$ applied to individual birds in the Caching task, plotted as in *Figure 4B*.

The online version of this article includes the following figure supplement(s) for figure 6:

**Figure supplement 1.** Chickadees return to cache locations even if seeds have been removed.

**Figure supplement 2.** Birds preferentially cache into unoccupied sites even when sites have not been recently checked.

neither model improved the fit to the data (*Figure 6F*). The best-fit values of $\tau_{occ}$ and $\tau_{emp}$ were very large (both >44 min, *Figure 6G*). Therefore, there did not appear to be a significant memory decay at the timescale of our experiments. This additionally implies that the preference for retrieving recaches may be a behavioral preference unrelated to a memory decay.

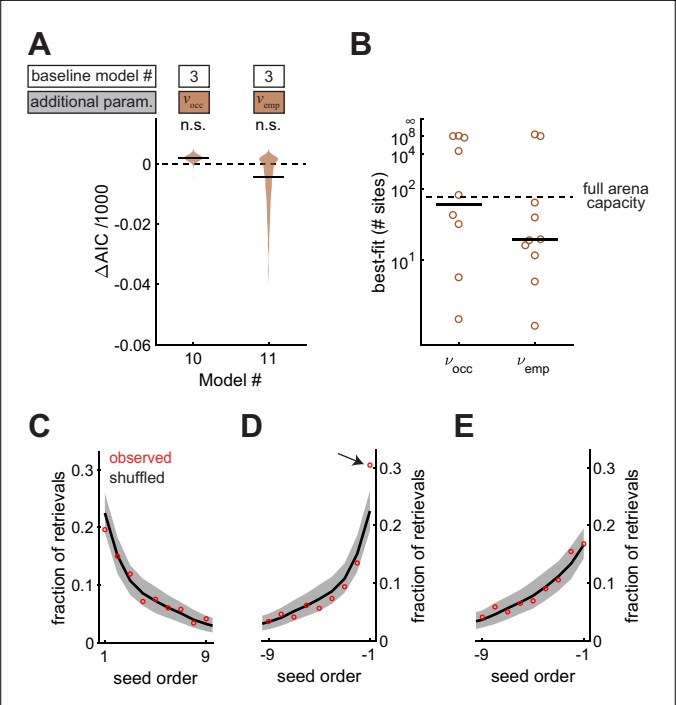

**Figure 7.** Memories are high capacity, and accessed in arbitrary order. (**A**) Performance of Models #10 and #11 applied to the Caching task, plotted as in *Figure 4A*. These models introduce free parameters $v_{occ}$ and $v_{emp}$ to quantify the decay of $\gamma_{occ}$ and $\gamma_{emp}$ as either the number of occupied sites or the number of checked-empty sites in the arena increases, respectively. (**B**) Best-fit values of $v_{occ}$ and $v_{emp}$ applied to individual birds in the Caching task, plotted as in *Figure 4B*. (**C**) Fraction of retrievals as a function of order in which seeds were caches. Order is aligned to the first cache. Red marker: mean observed fraction across birds. Black line: average value expected from retrieving seeds in a random order. Grey: 95% confidence interval of retrievals made in a random order. (**D**) Same as (**C**), but aligned to the most recent cache. Arrow indicates the only point that was significantly outside the 95% confidence interval of shuffled data. (**E**) Same as (**D**), but with transient caches eliminated. Transient caches are those that are retrieved without leaving the perch.

Chickadees often recheck sites, with a median of 2.4 min between consecutive checks of the same site. Our results suggest that these rechecks are not necessary for maintaining memory, since the memory decay in the above analysis is reset at every site check. To verify this independently of the model, we measured the fraction of caches that were made into empty sites. In this analysis, we only included sites that had not been recently checked by the bird. We found that chickadees prefer to cache into empty sites even after not checking sites for a long period of time (p < 0.001 after 30 min; *Figure 6—figure supplement 2*). Our results do not indicate whether memory remains stable over periods much longer than the behavioral session (1 hr), and whether rechecks become necessary at those timescales. However, prior work has showed some memories lasting up to a month, even when birds are not allowed to recheck their sites (*Roth et al., 2012*). Additionally even 1 hr delays are ethologically important for chickadees, since they typically retrieve their caches over the course of a single day in the wild (*Cowie et al., 1981*).

## Memories of site contents are of high-capacity and accessed in an arbitrary order

Food-caching birds are famous for storing and likely remembering large quantities of food items in the wild (*Pravosudov, 1985*). So far, it is unclear if chickadees exhibit a high memory capacity in our arena. For example, the statistical effects we observed can be explained by chickadees remembering a small number of caches and having no memory of other caches. To test this possibility, we allowed $\gamma_{occ}$ and $\gamma_{emp}$ to decay exponentially with the total number of occupied sites and checked-empty sites in the arena. The decay constants of these exponentials were two additional parameters, $v_{occ}$ and $v_{emp}$, respectively. As before, we fit two models (Model #10 and Model #11), each introducing one of

these two parameters. Neither model fit the data better than Model #3, in which $\gamma_{occ}$ and $\gamma_{emp}$ did not decay (*Figure 7A*). The best-fit values of $\nu_{occ}$ and $\nu_{emp}$ were >15 sites in both cases (*Figure 7B*) – comparable to the maximum number caches present in the arena at any single moment in the session (15.0 ± 1.0, N = 101 sessions). Therefore, memory did not appear to decay at capacities tested by our experiments.

All of our models so far have used spatial location to explain the choice of sites for caching and checking. We also asked whether the order in which seeds were cached was predictive of the order in which birds chose to retrieve them. For this analysis, we used the Caching task, in which birds often cached and retrieved large numbers of seeds. Across all retrievals, we asked how often the *n*th cached seed was chosen by the bird (*Figure 7C*). There was no difference between observed data and shuffled data, in which seeds were chosen randomly from the available caches. Thus, there was no detectable 'primacy' effect in the data (p = 0.96). We then asked how often the *n*th most recent seed was chosen. In this case, the most recently cached seed was retrieved more often than expected by chance (p < 0.001, *Figure 7D*). However, this 'recency' effect was largely caused by caches that the bird retrieved without leaving the site of the cache, which accounted for 14% of all retrievals (*Figure 7E*). We often observed chickadees making several of these 'transient caches' and retrievals with a single seed, before finally settling on a cache location. Once transient caches were excluded from the data, there was no detectable primacy or recency effect in the order of retrieval (p = 0.44 and p = 0.24, respectively).

## Discussion

We designed an apparatus to explore how chickadees make navigational decisions during food caching and retrieval. We built quantitative models to understand the relative contributions of mnemonic and non-mnemonic strategies to these behaviors. Our setup allowed birds to move within an arena similar to those used for rodent experiments (*Lever et al., 2002*; *Muller et al., 1987*; *Wilson and McNaughton, 1993*). We found that several natural features of chickadee spatial navigation were preserved in this environment. Our apparatus also allowed birds to cache and retrieve food from a relatively large number of concealed sites within the arena. Tracking behavior at these sites revealed that chickadees rely on multiple strategies for caching and retrieval and use memory in flexible ways during these behaviors.

We found two spatial strategies that did not require birds to remember the contents of individual sites: the use of spatial biases and a proximity effect. First, chickadees had spatial biases that were stable over time and unique to individual birds. Previous work has shown spatial biases in the wild. For example, individual birds can cache in different zones within a shared territory or on different parts of a tree (*Cowie et al., 1981*; *Dally et al., 2006*; *Lahti et al., 1998*). This strategy has a clear ethological advantage: biases reduce the memory load of cache locations, whereas choosing biases idiosyncratically minimizes unwanted overlap between individuals. The second contribution to behavior was the effect of proximity: chickadees tended to cache and check sites close to their previous site interaction. These effects have also been observed in the wild and have ethological advantages. Caching close to the food source minimizes travel distance and exposure to dangers (*Clarkson et al., 1986*; *Scott et al., 2017*; *Sherry et al., 2010*; *Stapanian and Smith, 1978*). Checking locations in close proximity is an efficient opportunistic method of exploring the environment (*Krebs et al., 1974*; *van der Vaart et al., 2011*). Although the patterns we find are on a dramatically smaller scale, it is intriguing to speculate that biases and proximity effects are driven by shared mechanisms across conditions.

We do not know whether these effects are truly non-mnemonic on timescales longer than our behavioral sessions. For example, a bird's experience over long periods of time may influence its biases or its typical trajectories within an environment (*Hampton and Sherry, 1994*; *Raby et al., 2007*). Regardless, explaining away biases and proximity effects was critical for isolating the remaining memory-guided components in our experiments.

Our analysis has revealed several features of chickadee memory. Memories were spatially specific, lasted the duration of the hour-long session, had high capacity, and could be retrieved in an order different from their storage order. These features have been studied before – usually in experiments specifically designed to look at each feature of memory in isolation. For example, radioactive tagging of food in the wild has shown that birds search for caches with centimeter precision (*Cowie et al., 1981*). In both field and laboratory experiments, birds continue to remember locations on both

hour-long timescales (*Clayton and Krebs, 1994b*; *Cowie et al., 1981*; *Sherry and Vaccarino, 1989*) and at least a month later (*Ekman et al., 1996*; *Roth et al., 2012*). Chickadees can cache thousands of seeds per day and are suspected to remember a large fraction of them (*Brodin, 2010*; *Pravosudov, 1985*). Finally, previous studies have also not detected a relationship between the order of caching and the order of retrieval, suggesting that chickadees have independent memories of caches rather than a single memory of the action sequence (*Balda and Kamil, 1989*; *Sherry, 1984a*; *Sherry, 1984b*). Our models capture these features on smaller spatial and temporal scales, and notably combine them within a single laboratory paradigm.

Classic work has shown that food-caching birds use memory to retrieve food (*Krushinskaya, 1966*; *Sherry, 1984a*; *Clayton and Dickinson, 1998*). They can even remember the contents of individual sites – for example, to choose caches that contain a certain type of food (*Clayton and Dickinson, 1998*; *Sherry, 1984a*). We also observed memories of site contents in our arena. Chickadees were attracted to occupied sites during retrieval, but avoided empty sites that they had previously checked. These results show that memories of site contents in chickadees not only affect a single behavior, but even influence the choice between different behaviors – in our case, site attraction or avoidance.

Site content influenced not only retrieval, but also caching itself. When caching, birds avoided occupied sites, but preferred sites that they had checked and found to be empty. These behaviors resulted in a 'spreading' of caches throughout an environment. Cache spreading is observed on a larger scale in the wild and is thought to be a defensive strategy against pilferers (*Alpern et al., 2012*; *Cowie et al., 1981*). However, it was not previously known that memory plays a role in this behavior. The implication of our results is that a single memory of a cached seed can be used by the bird for entirely different behaviors depending on context – in our case, caching or retrieval.

It has been shown that food-caching birds use memory in flexible ways. For example, they choose caches with certain types of food instead of others depending on selective satiety (*Clayton and Dickinson, 1999*) or knowledge about food perishability (*Clayton and Dickinson, 1998*; *Feeney et al., 2011*). Our results further show that memories might even be used for actions that are unrelated to immediate feeding needs – in our case to spread caches throughout an environment. The ability of the same memory to flexibly drive different behaviors that have different goals is a hallmark of episodic memories in humans (*Allen and Fortin, 2013*; *Clayton et al., 2003*; *Tolman, 1948*; *Tulving, 1972*). Our findings lends support to the idea that memories of food-caching birds are similarly general-purpose in nature.

## Materials and methods

### Resource availability

Lead contact

Further information and requests for resources and reagents should be directed to and will be fulfilled by the Lead Contact, Dmitriy Aronov (da2006@columbia.edu).

### Experimental model and subject details

All animal procedures were approved by the Columbia University Institutional Animal Care and Use Committee and carried out in accordance with the US National Institutes of Health guidelines. The subjects were 21 black-capped chickadees (*Poecile atricapillus*) collected from multiple sites in New York State using Federal and State scientific collection licenses. Subjects were at least 4 months old at the time of the experiment, but age was not determined more precisely. Chickadees are not visibly sexually dimorphic, and all experiments were performed blindly to sex. Sex was only determined on birds that were subsequently used for other procedures in the lab. For the Caching task, 17 chickadees were used (seven male, 3 female, 7 unknown). Of these birds, 7 made fewer than 64 caches across all sessions and were excluded from all analyses except those in *Figure 5—figure supplement 1A,C* (see reasoning in *Caching task* below). Of the remaining birds, six were subsequently used in the Retrieval task and four were used in experiments not described in this paper. One additional bird (sex unknown) was used in both the Caching and the Retrieval task, but the data for the Caching task was not included due to technical problems.

For the Retrieval task, 7 chickadees were used, 6 of which had been previously used in the Caching task (three male, 1 female, 3 unknown). The olfactory control experiment (*Figure 6—figure*

*supplement 1*) was performed on three birds, none of which were used in other tasks (sex unknown). These birds could not be used for any further experiments because they experienced their caches being removed from the experimental arena, which has been shown in prior literature to affect subsequent caching (*Hampton and Sherry, 1994*). Prior to experiments, all birds were housed in groups of 1–4 on a 'winter' light cycle (9 h:15 hr light:dark).

## Behavioral apparatus

### Design of caching apparatus

All custom parts of the behavioral arena were designed in Autodesk Inventor. Part files are available upon request. Two identical enclosed square arenas were constructed and used for all experiments. Each arena was a 61 cm x 61 cm square and contained 64 perches, 64 'cache sites', and 4 'feeder sites'.

The arena was constructed from five laser-cut layers. The top surface layer was a white polystyrene sheet (1.6 mm thick, McMaster-Carr, Atlanta, GA, USA, 8734K32). The second layer was a sheet of silicone rubber (0.8 mm thick, 60 A durometer, McMaster-Carr 1460N41). The third and fourth layers were black cast acrylic sheets (3.2 mm and 4.8 mm thick, McMaster-Carr, 8505K742; 8505K748, respectively). Finally, the fifth (bottom-most) layer of the arena was a transparent cast acrylic sheet (3.2 mm thick, McMaster-Carr 8560K257).

All five layers contained cutouts for the perches. These included space directly in front of and behind the perch, allowing birds to wrap their toes around the perch. The fourth layer included small ledges on which the ends of the perch rested. Perches were 9.5 mm diameter, 50.8 mm length wooden dowels (McMaster-Carr, 97195A434).

Cache sites were created by making square cutouts in the third and fourth layer. These cutouts formed a cavity 6.4 mm wide x 4.8 mm long x 7.9 mm deep. A 3D-printed ramp was inserted into this cavity and helped direct seeds into the bottom of the site (Clear resin, Formlabs, Somerville, MA, USA, RS-F2-GPCL-04). The bottom (fifth) layer contained no cutouts at the cache sites and therefore provided a floor for seeds to rest on. The second (silicone rubber) layer was cut on three sides above each cache site, creating a soft flap that could be pulled open. The top layer had a cutout above each site, providing access to the silicone flap. Finally, a white 3D-printed cap was attached to the top level next to each site (White Resin, Formlabs RS-F2-GPWH-04). This cap created a barrier that blocked visual access of site contents from anywhere in the arena.

The top four layers were screwed together. The bottom layer was attached using small magnets (K&J Magnetics, Pipersville, PA, USA, D22), and could be temporarily removed in order to clean out all cache sites between sessions.

Each feeder consisted of a 3D-printed basket (White Resin, Formlabs RS-F2-GPWH-04) attached to a stepper motor (Mouser, Mansfield, TX, USA, 108990003) that could open or close the feeder. The motor was controlled by an Arduino (Mouser, 713–102990189) via a stepper motor driver (Mouser, 474-ROB-12779). A ledge surrounding each feeder was 3D printed from colored PLA using a Taz 6 3D printer (Lulzbot, Fargo, ND, USA). Each feeder's ledge was a different color (red, blue, green, and yellow). Lights in the arena were turned on and off using a solid-state relay (Mouser, 558-D1D12), also controlled by the Arduino.

Each arena was placed inside a black plastic box with doors on one side. Walls of the box were positioned 2.5 cm from the edge of an arena, creating a 'moat' that almost completely surrounded the arena (except the corners). The box was 51 high. Bright shapes ~ 15 cm across (blue star, orange triangle, purple circle, and green tree) were positioned on the center of each wall. The arena was illuminated both from above and from below, using LED lights (Super Bright LEDs, St. Louis, MO, USA, VTL -x1515; NFLS-WW300X2-LC2). Natural sound of rushing water was played in the background to mask inadvertent room noises.

Birds were monitored using three cameras. A camera in the center of the ceiling was used to track bird's position (Amazon, Seattle, WA, USA, 180 degree Fisheye lens 1,080 p Wide Angle Pc Web USB camera, 30 fps). A camera positioned 90 cm below the arena (bottom camera) monitored the contents of cache sites and the bird's site checks (Edmund Optics, Barrington, NJ, USA, acA2500-60uc; #63–245, 1" python 5000 CMOS sensor, 57 fps). A third camera mounted in the center of one wall 24 cm above the floor of the arena (side camera) was used for manual verification of bird's behavior (Edmund Optics, EO-2323; #62–274, 2/3" Progressive Scan CMOS sensor, 50 fps).

## Actuation of the behavioral arena

For the Caching task, where no closed-loop manipulations were needed, all actuation (control of lights and motors) was performed by standalone Arduino code. For the Retrieval task, all task-relevant information (contents of cache sites and the bird's location) were monitored in real time by software written in Bonsai (*Lopes et al., 2015*). This software communicated with the Arduino to actuate events in the arena. Bonsai received real-time inputs from the bottom camera and the top camera. All codes are available upon request.

To detect occupied sites, an ROI was drawn around each cache site in the green channel of the video recorded by the bottom camera. The green channel was used because it provided high contrast with the red silicone material. At the beginning of each trial, the background of each ROI was subtracted using *BackgroundSubtraction* (subtractionMethod Absolute, ThresholdType ToZero, ThresholdValue 6). In each frame, novel objects were then detected using *FindContours* (Method ChainApproxNone, minArea 4) and counted using *BinaryRegionAnalysis*. The number of occupied sites was then passed through a lowpass filter ($y_n = 0.01x_n + 0.99y_{n-1}$, where $x_n$ is the count and $y_n$ is the filtered count). The value of $y_n$ rounded to the nearest integer was used by Bonsai as the current number of seeds in the arena. This filter introduced a delay of ~1 s between a cache occurring and being detected, ensuring that checks were not counted as caches.

To detect visits to feeders, an ROI was drawn around each feeder in the video recorded by the top camera. The sum of pixel values in this ROI was computed, and a threshold was chosen separately for each feeder to reliably detect the chickadee's head entering the ROI. Because chickadees are black-capped, all feeders were lighter in color, and ROI entrances corresponded to negative threshold crossing.

## Behavioral protocol

### Habituation

Birds selected for experiments were singly housed and weighed daily. Primary feathers were trimmed to prevent flight. Initially, birds were given an ad-libitum supply of small bird diet (Mazuri, St. Louis, MO, USA). Due to changes in animal husbandry protocols, for some birds, the diet was supplemented with dried mealworms. This change in diet had no significant effect on any results reported in the paper. Each bird's cage contained a small platform with 4–6 'replica' cache sites identical to those later used in the behavioral arena.

Birds were first given a 4-day period of acclimation and habituation to handling. After 4 days, they were gradually habituated to food restriction. First, birds were not given food for the first two hours of the lights-on period. This period was gradually increased to 4 hr (occasionally 3–5). Birds were weighed at the end of the food restriction period, and the length of this period was increased only if the bird's weight remained stable (fluctuations less than ~0.5 g) for 4 days. Experiments began after 14–21 days, once birds had stable weight on 3–5 hr of food restriction and were observed caching into the replica cache sites in their home cages. Some birds underwent 1–3 days of sessions in the olfactory control experiment (see *Experimental model and subject details*). Other birds underwent several weeks (14–45 days, median 25) of sessions in the Caching task and/or several weeks (7–82 days, median 58) of sessions in Retrieval task. For birds that were used in both of these tasks, the Caching task preceded the Retrieval task. In all tasks, prior to beginning the session, cache sites were emptied between sessions to 'reset' the arena.

### Caching task

Each bird was recorded in one 1-hr long behavioral session per day, starting immediately after the food restriction period. These sessions were run every day for the duration of the experiment. During these session, all four feeders in the arena were stocked with chopped sunflower seeds (~25 mg fragments). At any moment in time, the arena could be in one of five states. In state 0, all feeders were closed. In states 1–4, the corresponding feeder (1-4) was open, while the other three feeders were closed. In most sessions, the state of the arena at the beginning of the session was chosen randomly with probability 1/4 of choosing state 0 and 3/16 of choosing any of the other states. A timer was also started, with duration chosen randomly from an exponential distribution between 60 and 90 s with 15 s decay. Once the timer duration elapsed, a new state was chosen, with the same probabilities: 1/4 for state 0 and 3/16 for the others. The timer was restarted with a new randomly chosen duration. A

given state could repeat more than once, with one exception: if state 0 has occurred three times in a row, states 1–4 were chosen randomly with 1/4 probability each. In some early sessions, state 0 was omitted. In these sessions, the arena could be in states 1–4, and all transition probabilities between states were 1/4. For some earlier birds, the timer duration was also chosen from smaller values: exponential distribution between 30 and 60 s with 15 s decay. In later birds, this duration was switched to 60–90 s because it was observed to promote caching. There was no difference in parameters between birds run on either protocol.

In most session, we also turned lights off periodically, which helped motivate food seeking and caching behaviors. In these cases, 5 min lights-on periods were alternated with lights-off periods, which lasted 0.5, 1, or 2 min in different sessions. In early sessions for each bird, the lights-off period was omitted.

Except for the analysis in *Figure 5—figure supplement 1A,C*, we only analyzed data from those 10 birds that made at least 64 caches across all sessions. This threshold was chosen conservatively as the minimum number that could allow the bird to cache at least once into each site. For the analysis in *Figure 5—figure supplement 1A,C*, all 17 birds were used. For all analyses, we only included sessions where birds made at least 10 caches. For each bird, sessions were pooled across all feeder opening/closing and lights on/off conditions that were used for that bird.

## Retrieval task

Each bird was recorded in one behavioral session per day, which lasted at least 30 min (see below). These sessions were also run every day for the duration of the experiment. The session consisted of trials, each of which contained three phases: the feeder-open phase, the delay phase, and the feeder-closed phase.

The feeder-open phase consisted of two periods: the 'eating' period and the 'caching' period. In the eating period, one of the feeders was open, while the other three were closed. A timer was started at the beginning of this period, with a random duration chosen from an exponential distribution between 30 and 60 s with 15 s decay. Once the timer had elapsed, a new feeder was chosen to be open with 1/4 probability, with the other three feeders closed; the same feeder could be repeated more than once. The timer was restarted with a new randomly chosen duration.

The caching period started whenever the bird cached the first seed – that is, making one of the sites in the arena occupied. When this first cache was made, a new feeder was immediately chosen to be open with 1/4 probability. Subsequently, a new feeder was chosen with 1/4 probability whenever a new site was occupied by a cache, allowing the same feeder to be repeated more than once. This was done to motivate caching throughout the arena, rather than in the vicinity of the same feeder. When the bird visited the open feeder after caching the first seed, a 15 s timer was started. The feeder-open phase of the trial was terminated whenever this timer elapsed or three sites in the arena became occupied, whichever came first.

In the delay phase, all four feeders were closed, and lights in the arena were turned off. The delay phase lasted 2 min. After this 2 min period, lights were turned back on, and the feeder-closed phase was initiated.

In the feeder-closed phase, all four feeders remained closed at all times. If the bird retrieved all of the caches that it made earlier in the same trial, the feeder-closed phase terminated 15 sec after the removal of the last cache. This 15 s window was implemented to accommodate possible recaching of the last retrieved seed. The feeder-closed phase also terminated after 10 min if the bird had not retrieved all of the caches that were made in the same trial. After the feeder-closed phase terminated, a new trial was immediately started if the duration of the session had not yet exceeded 30 min. The session was also terminated after 30 min if the bird never made any caches.

## Experiment controlling for olfactory cues

Birds were recorded in three sessions per day. During the first session (habituation session), all feeders were closed, and the bird had no access to food. The bird's location in the arena was tracked for 5 min to measure the 'baseline' location distribution. Next, the bird was removed from the arena, then immediately reintroduced for another session (caching session). In the caching session, all feeders were open and the bird was allowed to make three caches. After three caches were made, the bird was again removed from the arena and placed into a cage with no food for a 30–60 min delay period.

During this delay, the feeders were closed and all food was removed from the arena, including the bird's caches and any scattered food. After the delay, the bird was reintroduced into the arena for the final session (test session). During this session, the bird's location was again tracked for 5 min.

## Quantification and statistical analysis

### Annotating site interactions

Using the bottom camera, we defined an ROI around each cache site. Pixel values from the first 100 frames of the video, when all cache sites were empty, were averaged to obtain a 'baseline' image for each ROI. For each subsequent frame in the video, we then measured two values: the Pearson correlation with the baseline image ($R_b$), and the Pearson correlation with the previous frame ($R_p$). $R_b$ was normalized by subtracting the median and dividing by the standard deviation. $R_p$ was transformed by subtracting a moving average of 200 frames. The MATLAB *findpeaks* function was used to detect changes in correlation that occurred in either one of these values ($R_b$ : minPeakDistance 100, minPeakHeight 10; $R_p$ : minPeakDistance 100, minPeakHeight 0.3). All of these events were visualized with side and bottom cameras and manually classified as caches, retrievals, checks, or false positives. Any period of time where the lights of the arena were turned off was excluded from this analysis.

### Location tracking with deep neural networks

To determine the bird's location, we trained a deep neural network (*Mathis et al., 2018*; *Nath et al., 2019*) to track two locations on the bird's body using the top camera: the tip of the beak and the location halfway between the two feet. We then defined an ROI around each feeder, as well as around the perch adjacent to each cache site. A feeder visit was defined as the bird's beak entering the corresponding ROI while the feeder was open. A cache site visit was defined as the bird's feet entering the corresponding ROI.

### Spatial distribution of caches and checks

Spatial distributions of caches (Caching task) and checks (Retrieval task) were quantified by computing the $p^{bias}$ values (see Probabilistic model of behavior). To calculate whether these distributions differed from a uniform distribution, we measured the entropy of $p^{bias}$ as

$$H = -\sum_i p_i^{bias} \log_2 p_i^{bias} \tag{1}$$

For our arena, $H$ could range between 0 bits for a maximally biased bird that uses only one site to six bits for perfectly uniform behavior. To assess statistical significance, we compared this value of $H$ to values from 10,000 simulations, in which the same number of events was drawn randomly from a uniform distribution spanning all sites in the arena.

To determine whether $p^{bias}$ was specific to individual birds, we divided each bird's sessions into two groups (A and B) and assigned each session to one of these groups. The two groups were equal-sized if the number of sessions was even, or differed by one session if the number of sessions was odd. We computed distributions of cache or check locations separately for the two groups ($p^{bias,A}$ and $p^{bias,B}$), using the same formula as for $p^{bias}$. We then computed the Pearson correlation between $p^{bias,A}$ and $p^{bias,B}$ for each bird and defined 'within-bird correlation' as the median of these values across birds. To compute 'across-bird correlations', we shuffled the identities of all birds 1000 times and computed the median Pearson correlation between $p^{bias,A}$ and $p^{bias,B}$ across these 1000 shuffles. The entire process was repeated for 1000 shuffled datasets, in which the assignment into groups A and B was randomized.

For the 'within PC1 cluster' analyses in *Figure 2D* and *Figure 2—figure supplement 2D*, we smoothed all values of $p^{bias}$ with a 3 × 3 site Gaussian window ($\sigma = 7.1$ cm, i.e. the distance between neighboring sites). We computed the first principal component of all smoothed $p^{bias}$ values across birds. We grouped all birds into two clusters, using k-means clustering. We then performed the same process as above, but with one exception: during the shuffling of bird identities, a bird was only allowed the identity of a bird from the same cluster.

To measure the stability of $p^{bias}$ over time, we measured the distribution of cache or check locations for each trial separately, using the same formula as for $p^{bias}$. We smoothed these distributions with a 3 × 3 site Gaussian window ($\sigma = 7.1$ cm). We then measured the Pearson correlation value across all

pairs of smoothed distributions that were $\Delta s$ sessions apart and took the median across all of these pairs of sessions. This value was computed for all $\Delta s$ from one to $N$. Here, $N$ was determined as the value for which in the Caching task there were at least 2 pairs of sessions available from at least five of the birds. This value was $N = 8$ and was applied to both tasks.

## Statistics

Unless otherwise stated, the following statistical methods were used throughout the paper. To test whether a given model (Model A) was a significant improvement over another model (Model B), we computed AIC values for the two models and the difference between these values (ΔAIC). A lower AIC value for Model A (ΔAIC < 0) would indicate its improvement over Model B. To determine the statistical significance of such a result, we recomputed ΔAIC for 1000 samples bootstrapped across birds. We then found the fraction of these bootstrapped ΔAIC values that were >0 and reported that fraction as the p-value. These values are summarized in *Table 1*.

For indicating the best-fit parameter values, we reported the median and the standard error of the median across birds. To test whether these values were significantly different from 0 across all birds, we used a one-tailed Wilcoxon signed rank test. To determine whether parameter values were individually significant for a particular bird, we bootstrapped the behavioral sessions of that bird and re-fit the model to each set of bootstrapped data. We then measured the p-value the fraction of bootstrapped parameters that were above or below 0 (depending on the direction of the effect across birds). These values are summarized in *Table 2*.

## Probabilistic model of behavior

Consider a bird recorded in a set of behavioral sessions within the arena. This arena contains multiple discrete *sites*. Each site is either a *cache site* (i.e. location covered by a silicone flap) or a *feeder*. We will consider events that occur at these sites.

We first consider cache sites. At a cache site, a *check* is an opening of the silicone rubber flap by the bird, which allows the bird visual access to the contents of the site. A *cache* is a placement of seeds into the site. A *retrieval* is a removal of seeds from the site. Caches and retrievals necessarily require the bird to open the silicone flap; therefore, each cache and retrieval at a cache site is also considered to be a check. For feeders, we define a *retrieval* as a removal of a seed. A retrieval is the only type of an event that is considered at feeders.

We define a site *interaction* as any cache, check, or retrieval – either at a cache site or at a feeder. Visits to sites that do not include at least one of these three types of events (e.g. landing at a site) are not considered to be interactions and are not included in the analysis.

For each bird, the experiment consists of non-overlapping *trials*. In the Caching task, each behavioral session is considered to be one trial. In the Retrieval task, each trial is a period of time consisting of the *feeder-open* phase, the *delay* phase, and the *feeder-closed* phase. In this task, each behavioral session contains at least one trial, and usually multiple trials.

Each site is considered to be *occupied* if it contains at least one seed and *empty* if it does not. Feeders have a large capacity and are occupied at all times. Cache sites are considered to be empty at the beginning of each trial. If some cache sites contain seeds remaining from a previous trial, those seeds are ignored in the analysis. This occasionally happens in the Retrieval task if the bird had failed to retrieve all the caches during one of the previous trials (< 1% of all retrievals). Any withdrawal of such a seed is not considered to be a retrieval. However, any subsequent placement of that seed into a cache site is considered to be a cache.

## Definitions of variables

The following variables are defined for each bird:

> $S$ is the number of sites in the arena. In the arena described in this paper, there are 64 cache sites and four feeders. Therefore, $S = 68$.
> $R$ is the number of behavioral trials.
> $I$ is the number of site interactions performed by the bird across all $R$ trials.

For each site interaction $i$, where $1 \leq i \leq I$, the following are defined:

$s_i$ is the index of the site where the interaction occurred, where $1 \leq s_i \leq S$.

$r_i$ is the index of the trial during which the interaction occurred, where $1 \leq r_i \leq R$.

$\phi_i$ is the phase of the trial during which the interaction occurred, defined as $\phi_i = 0$ for all site interactions during the Caching task, $\phi_i = 1$ for interactions during the feeder-open phase of the Retrieval task, and $\phi_i = 2$ for interactions during the feeder-closed phase of the Retrieval task. There are no interactions during the delay phase of the Retrieval task.

$\Delta_i$ is the change in the number of seeds at site $s_i$ during the interaction. I.e., $\Delta_i < 0$ for retrievals, $\Delta_i > 0$ for caches, and $\Delta_i = 0$ for checks that are not coincident with either a cache or a retrieval.

For each site $s$, where $1 \leq s_i \leq S$, the following is defined:

$\vec{x}_s$ is the location of the site. Here, $\vec{x}_s$ is a vector indicating location in 2-dimensional space. Several additional variables are defined for convenience of notation.

The Euclidean distance between sites $s$ and $s'$ is:

$$d_{ss'} = \left\| \vec{x}_s - \vec{x}_{s'} \right\| \tag{2}$$

Next, we consider the time point immediately preceding interaction $i$. Several variables are defined to indicate the state of the behavioral arena at this point. For the following definitions, 'current trial' is the trial that contains interaction $i$.

$n_{is}$ is the occupancy of site $s$. The value is $n_{is} = 1$ if the site is occupied, or $n_{is} = 0$ if the site is empty.

$N_i$ is the total occupancy of the arena – that is, the number of cache sites that are occupied.

$t_{is}$ is the amount of time that has elapsed since the most recent interaction with site $s$ in the current trial. If the bird has not interacted with site $s$ in the current trial, then $t_{is} = \infty$.

$c_{is}$ indicates whether site $s$ is 'checked-empty'. That is, $c_{is} = 1$ if the bird has checked site $s$ in the current trial, and that site is empty. If the bird has not checked site $s$ in the current trial, or if site $s$ is occupied, then $c_{is} = 0$.

$C_i$ is the number of checked-empty cache sites in the arena.

$n_{is}^{\mathrm{r}}$ indicates whether the latest cache into site $s$ in the current trial has been a recache. That is, $n_{is}^{\mathrm{r}} = 1$ if the bird has cached into site $s$ during the feeder-closed phase of the task, and site $s$ is occupied. If the bird has only cached into site $s$ during the feeder-open phase, or if site $s$ is empty, then $n_{is}^{\mathrm{r}} = 0$.

$\mu_i$ indicates whether the next interaction in the current trial that involves a seed is a cache. If the next interaction involving a seed is a cache, $\mu_i = 1$. If the next such interaction is a retrieval, or if the bird never interacts with a seed again in the current trial, $\mu_i = 0$.

**Table 3.** Masks for different subsets of interactions.

| Subset of interactions | $G_i$ | $H_i$ |
|---|---|---|
| All caches in the Caching task | $G_i = \begin{cases} 1, & \phi_i = 0, \Delta_i > 0 \\ 0, & \text{otherwise} \end{cases}$ | $H_i = \begin{cases} 1, & \phi_i = 0, \Delta_i > 0 \\ 0, & \text{otherwise} \end{cases}$ |
| First checks in the Caching task. This mask includes the first check that the bird made after retrieving each seed that was subsequently cached (i.e. it does not include checks made after retrieving seeds that were subsequently eaten without caching). | $G_i = \begin{cases} 1, & \phi_i = 0, \Delta_i > 0 \\ 0, & \text{otherwise} \end{cases}$ | $H_i = \begin{cases} 1, & \phi_i = 0, \Delta_{i-1} < 0, \mu_i = 1 \\ 0, & \text{otherwise} \end{cases}$ |
| All checks in the feeder-closed phase of the Retrieval task | $G_i = \begin{cases} 1, & \phi_i = 2 \\ 0, & \text{otherwise} \end{cases}$ | $H_i = \begin{cases} 1, & \phi_i = 2 \\ 0, & \text{otherwise} \end{cases}$ |
| All checks up to and including finding a cache (i.e. making the first check of an occupied site) in the feeder-closed phase of the Retrieval task. If no cache is found in the feeder-closed phase of the trial, this includes all checks up until the end of the trial. | $G_i = \begin{cases} 1, & \phi_i = 2 \\ 0, & \text{otherwise} \end{cases}$ | $H_i = \begin{cases} 1, & \phi_i = 2, \sum\limits_{\substack{j<i \\ r_j=r_i}} n_{js_j} = 0 \\ 0, & \text{otherwise} \end{cases}$ |
| First checks in the feeder-closed phase of the Retrieval task. This mask includes the first check made during the feeder-closed phase of each trial. | $G_i = \begin{cases} 1, & \phi_i = 2 \\ 0, & \text{otherwise} \end{cases}$ | $H_{ik} = \begin{cases} 1, & \phi_{i-1} = 1, \phi_i = 2 \\ 0, & \text{otherwise} \end{cases}$ |

## Model parameters

Twelve parameters are used by various models described in the main text. These are defined in the factors described in *Equations 5–9*. We define a vector of these parameters:

$$\theta = \left[ \begin{array}{cccccccccccc} \gamma_{\mathrm{prv}} & \sigma_{\mathrm{prv}} & \gamma_{\mathrm{occ}} & \sigma_{\mathrm{occ}} & \gamma_{\mathrm{emp}} & \sigma_{\mathrm{emp}} & \gamma_{\mathrm{occ}}^{\mathrm{c}} & \gamma_{\mathrm{occ}}^{\mathrm{r}} & 1/\tau_{\mathrm{occ}} & 1/\tau_{\mathrm{emp}} & 1/\nu_{\mathrm{occ}} & 1/\nu_{\mathrm{emp}} \end{array} \right] \tag{3}$$

Inversions of the last four parameters are done for the purpose of parameter regularization, described later.

## Interaction masks

Models described in the text are applied to specific subsets of interactions. A single model is sometimes applied to different subsets: for example, Model #1 is used to quantify the effect of proximity either on caches or on checks, which are two different subsets of interactions. To define such subsets, we use *interaction masks* (*Table 3*). For each interaction $i$, the value of the mask is 1 if that interaction is included in the analysis or 0 if it is not. For different formulas below, two masks will be used: $G_i$ is used to mask interactions based on the task (Caching or Retrieval); $H_i$ is used to further mask subsets of interactions within these tasks.

## Scaling factors

In the baseline model (Model #0), the probability of an interaction occurring at a particular site given by the distribution $p^{\mathrm{bias}}$. We compute $p^{\mathrm{bias}}$ for either the Caching task or the Retrieval task using the mask $G$ described above. For every site $s$:

$$p_s^{\mathrm{bias}} = \frac{\sum_{i=1}^{I} G_i \delta_{s_i s}}{\sum_{i=1}^{I} G_i} \tag{4}$$

where $\delta$ is the Kronecker Delta, defined as $\delta_{xy} = 1$ if $x = y$, and $\delta_{xy} = 0$ if $x \neq y$.

We next define *scaling factors*. A scaling factor is a function defined for each interaction $i$ and site $s$. These factors will be used later to scale values of $p^{\mathrm{bias}}$. Positive values of these scaling factors indicate an increase in probability, and negative values indicate a decrease in probability.

The first factor is used to model the effect proximity to the previous site interaction. It is a Gaussian that decays with distance from the previous site, defined at distances > 0. Empirically, we found that models fit better if the value of this scaling factor was set to 0 at a distance of 0. This is due to the fact that birds were unlikely to interact with the exact same site twice in a row, even though they were highly likely to interact with neighboring site. Therefore, we defined the scaling factor as follows:

$$\alpha_{is}^{(1)} = \begin{cases} \gamma_{\mathrm{prv}} e^{-d_{s_{i-1} s}^2 / 2\sigma_{\mathrm{prv}}^2}, & s \neq s_{i-1} \\ 0, & \text{otherwise} \end{cases} \tag{5}$$

The second factor is used to model the effect of distance from occupied sites in the arena. It is a summation of Gaussians, each centered at an occupied site. The amplitude decays in time (in order to model memory decay) and with the occupancy of the arena (in order to model memory capacity). Note that in most models in the paper these decays are not included (i.e. $\tau_{\mathrm{occ}} = \infty$ and/or $\nu_{\mathrm{occ}} = \infty$).

$$\alpha_{is}^{(2)} = \gamma_{\mathrm{occ}} \sum_{s'=1}^{S} \left( e^{-d_{ss'}^2 / 2\sigma_{\mathrm{occ}}^2} \right) \left( e^{-t_{is'} / \tau_{\mathrm{occ}}} \right) \left( e^{-N_i / \nu_{\mathrm{occ}}} \right) n_{is'} \tag{6}$$

The third factor is analogous to the second factor, but for checked-empty sites in the arena:

$$\alpha_{is}^{(3)} = \gamma_{\mathrm{emp}} \sum_{s'=1}^{S} \left( e^{-d_{ss'}^2 / 2\sigma_{\mathrm{emp}}^2} \right) \left( e^{-t_{is'} / \tau_{\mathrm{emp}}} \right) \left( e^{-C_i / \nu_{\mathrm{emp}}} \right) c_{is'} \tag{7}$$

The fourth and fifth factors are used to separately model the effects of caches made in the feeder-open phase of the trial and caches made in the feeder-closed phase (i.e. 'recaches'):

$$\alpha_{is}^{(4)} = \gamma_{\mathrm{occ}}^{\mathrm{r}} n_{is}^{\mathrm{r}} \tag{8}$$

**Table 4.** Permitted ranges of parameters, $P(M, z)$, for main models.

| | Parameter | | | | | | | | | | | | |
|---|---|---|---|---|---|---|---|---|---|---|---|---|---|
| | $\gamma_{\text{prv}}$ | $\sigma_{\text{prv}}$ | $\gamma_{\text{occ}}$ | $\sigma_{\text{occ}}$ | $\gamma_{\text{emp}}$ | $\sigma_{\text{emp}}$ | $\gamma_{\text{occ}}^{\text{c}}$ | $\gamma_{\text{occ}}^{\text{r}}$ | $1/\tau_{\text{occ}}$ | $1/\tau_{\text{emp}}$ | $1/\nu_{\text{occ}}$ | $1/\nu_{\text{emp}}$ | |
| Model index | | | | | | Parameter index ($z$) | | | | | | | # free param |
| (M) | 1 | 2 | 3 | 4 | 5 | 6 | 7 | 8 | 9 | 10 | 11 | 12 | ($f_{\text{M}}$) |
| 0 | {0} | {0} | {0} | {0} | {0} | {0} | {0} | {0} | {0} | {0} | {0} | {0} | 0 |
| 1 | $(-\infty,\infty)$ | $[0,\infty)$ | {0} | {0} | {0} | {0} | {0} | {0} | {0} | {0} | {0} | {0} | 2 |
| 2 | $(-\infty,\infty)$ | $[0,\infty)$ | $(-\infty,\infty)$ | {0} | {0} | {0} | {0} | {0} | {0} | {0} | {0} | {0} | 3 |
| 3 | $(-\infty,\infty)$ | $[0,\infty)$ | $(-\infty,\infty)$ | {0} | $(-\infty,\infty)$ | {0} | {0} | {0} | {0} | {0} | {0} | {0} | 4 |
| 4 | $(-\infty,\infty)$ | $[0,\infty)$ | $(-\infty,\infty)$ | $[0,\infty)$ | $(-\infty,\infty)$ | {0} | {0} | {0} | {0} | {0} | {0} | {0} | 5 |
| 5 | $(-\infty,\infty)$ | $[0,\infty)$ | $(-\infty,\infty)$ | {0} | $(-\infty,\infty)$ | $[0,\infty)$ | {0} | {0} | {0} | {0} | {0} | {0} | 5 |
| 6 | $(-\infty,\infty)$ | $[0,\infty)$ | {0} | {0} | {0} | {0} | $(-\infty,\infty)$ | {0} | {0} | {0} | {0} | {0} | 3 |
| 7 | $(-\infty,\infty)$ | $[0,\infty)$ | {0} | {0} | {0} | {0} | $(-\infty,\infty)$ | $(-\infty,\infty)$ | {0} | {0} | {0} | {0} | 4 |
| 8 | $(-\infty,\infty)$ | $[0,\infty)$ | $(-\infty,\infty)$ | {0} | $(-\infty,\infty)$ | {0} | {0} | {0} | $[0,\infty)$ | {0} | {0} | {0} | 5 |
| 9 | $(-\infty,\infty)$ | $[0,\infty)$ | $(-\infty,\infty)$ | {0} | $(-\infty,\infty)$ | {0} | {0} | {0} | {0} | $[0,\infty)$ | {0} | {0} | 5 |
| 10 | $(-\infty,\infty)$ | $[0,\infty)$ | $(-\infty,\infty)$ | {0} | $(-\infty,\infty)$ | {0} | {0} | {0} | {0} | {0} | $[0,\infty)$ | {0} | 5 |
| 11 | $(-\infty,\infty)$ | $[0,\infty)$ | $(-\infty,\infty)$ | {0} | $(-\infty,\infty)$ | {0} | {0} | {0} | {0} | {0} | {0} | $[0,\infty)$ | 5 |

$$\alpha_{is}^{(5)} = \gamma_{\text{occ}}^{\text{c}} \left( n_{is} - n_{is}^{\text{r}} \right) \tag{9}$$

Each of the scaling factors $k = 1, 2, \ldots, 5$ is converted to a multiplicative form as follows:

$$\beta_{is}^{(k)} = 10^{\alpha_{is}^{(k)}} \tag{10}$$

Un-normalized probability of interaction $i$ occurring at site $s$ is then computed by multiplying all five scaling factors by the baseline probability. Because this value depends on all model parameters, we denote it as a function of $\theta$:

$$p_{is} (\theta) = p_s^{\text{bias}} \prod_{k=1}^{5} \beta_{is}^{(k)} \tag{11}$$

This value is converted to normalized probability that sums to one across all sites of the arena:

$$\hat{p}_{is} (\theta) = \frac{p_{is}(\theta)}{\sum_{s'=1}^{S} p_{is'}(\theta)} \tag{12}$$

## Model definitions

We describe twelve different models in the main text. All of these models can be expressed using the mathematical formulation described above. The difference between these models is the parameter space: In some models, a particular parameter is fixed at a value of 0, whereas in other models, that same parameter is a free parameter permitted to take values within a certain range. *Table 4* specifies the parameter space for each model. Models are numbered using index $M = 0, 1, \ldots, 11$. Parameters within vector $\theta$ are numbered using index $z = 1, 2, \ldots, 12$. For each model $M$ and parameter $z$, the table indicates $P(M, z)$ – the set of permitted values for that parameter. For parameters whose values are fixed at zero, $P(M, z) = \{0\}$ . We use $f_{M}$ to indicate the number of free parameters in model $M$; this value is also given in the table.

For each model, the parameter space is defined as the Cartesian product of permitted ranges across all twelve parameters:

$$\Theta(M) = P(M, 1) \times P(M, 2) \times \ldots \times P(M, 12) \tag{13}$$

## Log-likelihood computation

The log-likelihood of the model is computed by adding log-probabilities across all interactions observed in the data. For this purpose, the interaction mask $H$ is used to only include the subset of the interactions that is being fit by the model:

$$\mathcal{L}(\theta) = \sum_{i=1}^{I} H_i \ln \hat{p}_{is_i}(\theta) \tag{14}$$

To pull data across birds, we denote the log-likelihood for each bird $b$ as $\mathcal{L}_b(\theta)$. Here $1 \leq b \leq B$, where $B$ is the number of birds. The pooled log-likelihood is then computed as:

$$\mathcal{L}^{\text{pool}}(\theta) = \sum_{b=1}^{B} \mathcal{L}_b(\theta) \tag{15}$$

The 'cost' of the model fit is computed by negating the likelihood and adding a Ridge regularization term. Ridge regularization penalizes large magnitudes of parameters and helps prevent overfitting. For bird $b$, the cost is:

$$J_b(\theta) = -\mathcal{L}_b(\theta) + \lambda \sum_{j=1}^{12} \theta_j^2 \tag{16}$$

For the pooled data, the cost is similarly:

$$J^{\text{pool}}(\theta) = -\mathcal{L}^{\text{pool}}(\theta) + \lambda \sum_{j=1}^{12} \theta_j^2 \tag{17}$$

Here, $\lambda$ is the regularization parameter. We use $\lambda = 1$ for the Caching task and $\lambda = 10$ for the Retrieval task. These values are different due to different numbers of data points in the two tasks.

## Model fitting and evaluation

To fit each model $M$ using maximum-likelihood estimation, we determine values within the parameter space of model $M$ that minimize the cost. These best-fit parameters are given by $\hat{\theta}_b(M)$ for each bird $b$ and by $\hat{\theta}^{\text{pool}}(M)$ for the pooled data:

$$\hat{\theta}_b(M) = \underset{\theta \in \Theta(M)}{\arg\min}\, J_b(\theta) \tag{18}$$

$$\hat{\theta}^{\text{pool}}(M) = \underset{\theta \in \Theta(M)}{\arg\min}\, J^{\text{pool}}(\theta) \tag{19}$$

We fit each model using the MATLAB *fmincon* function. Each model was fit five times with different initial conditions, and the solution with the minimum cost was selected.

To evaluate each model, we used the Akaike Information Criterion (AIC). Smaller AIC values indicate better model performance. AIC value is improved by increased model likelihood, but is penalized by adding extra parameters. We computed the AIC value on the pooled data:

$$\text{AIC}(M) = -2\mathcal{L}^{\text{pool}}(\hat{\theta}^{\text{pool}}(M)) + 2f_M \tag{20}$$

To determine significance, 1,000 pools of data were constructed by bootstrapping across birds. For each bootstrap, the AIC values of all models were calculated and compared to find the percentage of bootstraps where model performance improved.

# Acknowledgements

We thank D Scheck, S Hale, T Tabachnik, R Hormigo, and K Gutnichenko for technical assistance; the Black Rock Forest Consortium, J Scribner and the Hickory Hill Farm, and T Green for help with field work; L Abbott and members of the Aronov lab for comments on the manuscript. Illustrations in *Figure 1B* are by J Kuhl.

# Additional information

## Funding

| Funder | Grant reference number | Author |
|---|---|---|
| National Science Foundation | Graduate Research Fellowship Program | Marissa C Applegate |
| New York Stem Cell Foundation | Robertson Neuroscience Investigator Award | Dmitriy Aronov |
| National Institutes of Health | NIH Director's New Innovator Award (DP2-AG071918) | Dmitriy Aronov |
| Arnold and Mabel Beckman Foundation | Beckman Young Investigator Award | Dmitriy Aronov |
| National Institutes of Health | T32 EY013933 | Marissa C Applegate |

The funders had no role in study design, data collection and interpretation, or the decision to submit the work for publication.

## Author contributions

Marissa C Applegate, Conceptualization, Data curation, Formal analysis, Funding acquisition, Investigation, Methodology, Resources, Software, Validation, Visualization, Writing – original draft, Writing – review and editing; Dmitriy Aronov, Conceptualization, Formal analysis, Funding acquisition, Methodology, Resources, Supervision, Writing – original draft, Writing – review and editing

## Author ORCIDs

Marissa C Applegate http://orcid.org/0000-0002-3717-2477

## Ethics

All animal procedures were approved by the Columbia University Institutional Animal Care and Use Committee (Protocol AAAR5402) and carried out in accordance with the US National Institutes of Health guidelines. Wild birds were collected under Federal and NY State scientific collection licenses.

## Decision letter and Author response

Decision letter https://doi.org/10.7554/eLife.70600.sa1
Author response https://doi.org/10.7554/eLife.70600.sa2

# Additional files

## Supplementary files
• Transparent reporting form

## Data availability

The following data is located on Dryad: For the Caching task, we include annotations of all trials. For each trial, we annotate the time and location of every cache, retrieval, and site check, as well as the bird's location at every timepoint. We also include the times and identity of any feeder sites that were open, as well as the times of any lights-off periods. For the Retrieval task, we include annotations of all trials. For each trial, we annotate the location and time of any caches made during the feeder-open phase of the trial, as well as the time that feeder-closed phase began and ended. We also included the time and location of the bird's recaches, retrievals and site checks during the trial. We finally include the bird's location at every time point.For the control experiment described in Figure 6—figure supplement 1, we include annotations of the location of the bird for every timepoint in the habituation session and the test session, as well as the location of caches from the caching session. For all behavioral models described in the paper, we include the bootstrapped AIC values as well as the best-fit values of all model parameters for each bird.

The following dataset was generated:

| Author(s) | Year | Dataset title | Dataset URL | Database and Identifier |
|---|---|---|---|---|
| Applegate M, Aronov D | 2022 | Flexible use of memory by food-caching birds | https://doi.org/10.5061/dryad.6wwpzgn12 | Dryad Digital Repository, 10.5061/dryad.6wwpzgn12 |

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
