## [Editor Report]

The extreme memory capacities of food-caching birds provide untapped opportunities for studying mechanisms of memory formation and retrieval. Here, Applegate and Aronov develop an automated animal and cache-site tracking system in which moments of seed deposits, retrievals, and checks are measured continuously alongside the animal's spatial positions. Probabilistic models reveal idiosyncratic spatial preferences in individual birds and also identify flexible memory usage – in which a single memory of past seed deposition can differentially guide spatial trajectories depending on if the bird is in engaged in retrieving or storing seeds. The rigorous behavioral tracking and modeling sets the stage for dissection of neural mechanisms underlying memory storage and retrieval.

---

## [Decision Letter]

**Decision letter after peer review:**

Thank you for submitting your article "Flexible use of memory by food-caching birds" for consideration by *eLife*. Your article has been reviewed by 3 peer reviewers, one of whom is a member of our Board of Reviewing Editors, and the evaluation has been overseen by Laura Colgin as the Senior Editor. The reviewers have opted to remain anonymous.

Essential revisions:

1) All reviewers agreed that the paper overstates the novelty of the behavioral findings. Please address issues raised below.

A main concern is to what extent this paper truly offers an original behavioral result with respect to the flexibility of memory. It is known that birds can remember seed caches and the major gains of the present manuscript are not revealing a new faculty but rather describing a method by which to study that faculty in the laboratory. Also, the sample size is rather small (N = 7 to 17) and variable across figures and analyses. For example, many readers will be familiar with decades-old studies by Sherry, Clayton, and Shettleworth that showed similar flexible memory use by caching birds (e.g. Raby et al., 2007; Shettleworth, 1990; Clayton et al., 2003). As is, the current introduction simply states that flexible memory is 'a matter of debate' in animals – but those familiar with previous studies of caching would likely disagree. The authors should re-write the introduction and abstract to be as generous as possible to past field work, to specify precisely what was previously known, and what is the truly new behavioral finding (if any) in the present work. For example, it's conceivable that this paper could be framed differently, i.e. "it has long been known that caching birds exhibit flexible memory that enable them to target specific sites differentially for depositing versus retrieving seeds, here we leverage this flexible memory capacity to quantify precisely how remembered cache locations influence foraging on a moment to moment basis." (or something like this…)

As in, a potential strength of the paper is the behavioral modeling which identifies the precise candidate considerations (parameters) that influence moment by moment aspects of foraging. This model has the potential to provide an algorithmic level description of the task, which in turn provides numerical values that could conceivably be identified in neural signals in future recording studies (see point 2 below to shore up this potential). It's one thing to show that caching birds have flexible memory (as past work has shown), but what seems uniquely new about this paper is that it leverages this past work towards a model that makes instant by instant predictions for how 'special' places influence behavior in a space- and time-dependent way. This algorithmic level description of the behavior is what could be novel, which should therefore be emphasized in the abstract/intro.

(2) If, as the reviewers suggest, the focus of the paper shifts from a descriptive account of the behavior in the new preparation to an emphasis on the validity of the statistical models, then more needs to be done to motivate the modeling decisions. It is currently not convincing that the parameters of the model are dissociable, especially since birds can continually cache and retrieve and recache.

Here are some unanswered questions that should be addressed if we are to take the quantitative parameter estimates of the models seriously, rather than the overall goodness of fit revealing the existence of some feature of the behavior (e.g. behavior is spatially autocorrelated).

(2.1.) Is log-linear interaction of the parameters appropriate or are there more complex dependencies?

(2.2) Is a Gaussian spatial autocorrelation function appropriate? Is the standard deviation fixed across space? Across strategy?

(2.3) Is time-dependent forgetting tested/modeled appropriately? There was an odd mismatch between the statement about preference for re-caching and the lack of a forgetting curve. What is the longest the animals will have a benefit from remembering caches? If only during one session, is this then perhaps a weakness in the task since one would expect birds in the wild to cache for longer time periods? Given that birds are rechecking single sites every 2.4 minutes, the authors should provide a more convincing argument how this unstructured task addresses the issue of memory decay.

(3) Please clarify how the spatial biases identified for individual birds affect caching strategy.

Currently the authors separately present first that birds have individual spatial biases and later on with a modelling approach show that previous visits and general proximity also biases the animals search and caching behaviour. But how do the two aspects relate to each other? One would assume that a general individual bias would decrease the number of possible locations the animal in general has to look at and therefore it is perhaps a mix of memories being used: the bias memory as well as the episodic memory of caching food. If one would include the bias as well as the principles that the animals have a proximity preference, how much variance will still be explained by the episodic memory? Or is the individual bias a result of the proximity bias? It would be good if the authors could create one major model which includes all identified factors to see how much which factor contributes.

(4) Please provide supplemental videos and increased clarity of the experimental timeline.

Further, it would help readers to understand the details of the behaviour if for example videos of the bird behaviour are provided with the manuscript for the different types of trials. That would help the reader understand in which times scales the animal is remembering and deciding. Further, an overall schematic of the timeline for the birds would be helpful. How long were they trained and habituated on what? How many sessions per day? How many days/weeks in total? For example days/weeks could go to the right and then down would be time within a day since multiple sessions were done per day. Then it can also be marked which data was included for analysis.

(5) Several technical points, if addressed, could make the statistics a bit stronger.

First, for the description of spatial biases, it appears that the same data is used to assign cluster membership and then quantify cluster separation. Some sort of cross-validation should be used throughout the analyses plotted in Figure 2 (Panel 2E notwithstanding).

Second, for the model fitting, a point estimate is used. It would be good to know the confidence around these point estimate and here a Bayesian framework would be helpful perhaps using Markov chain Monte Carlo methods to finds highest density intervals for likely parameter estimates (e.g. Annis and Palmeri 2017, doi: 10.1002/wcs.1458; Kruschke 2013 DOI: 10.1037/a0029146).*Reviewer #1 (Recommendations for the authors):*

Applegate and Aronov develop an automated animal and cache-site tracking system to study foraging strategies in chickadees. A key strength of this system is that each moment of seed deposit, retrieval, and 'checks' is marked alongside the animal's spatial position. Tracking animals and their caching presents tremendous opportunities for the study of memory storage and retrieval. The paper's main strength is the quantitative modeling of the caching behavior, which is elegant and has explanatory power over moment-to-moment navigation decisions. The paper's main weakness is overstatement of the novelty of the discovery of flexible memory use by caching birds. This paper need not 'discover' flexibly memory use to be of general interest for *eLife* readership. Leveraging this very unique behavioral capacity for the generation of highly predictive foraging models is sufficient on its own to make a large impact.

(1) The setup of this paper unnecessarily overstates the novelty of the behavior and understates the utility of the behavioral modeling for future mechanistic studies.

A main concern is to what extent this paper truly offers an original behavioral result with respect to the flexibility of memory. For example, many readers will be familiar with decades-old studies by Sherry, Clayton, and Shettleworth that showed similar flexible memory use by caching birds (e.g. Raby et al., 2007; Shettleworth, 1990; Clayton et al., 2003). As is, the current introduction simply states that flexible memory is 'a matter of debate' in animals – but those familiar with previous studies of caching would likely disagree. The authors should re-write the introduction and abstract to be as generous as possible to past field work, to specify precisely what was previously known, and what is the truly new behavioral finding (if any) in the present work. For example, it's conceivable that this paper could be framed differently, i.e. "it has long been known that caching birds exhibit flexible memory that enable them to target specific sites differentially for depositing versus retrieving seeds, here we leverage this flexible memory capacity to quantify precisely how remembered cache locations influence foraging on a moment to moment basis." (or something like this…)

As in, the strength of the paper is the behavioral modeling which identifies the precise candidate considerations (parameters) that influence moment by moment aspects of foraging. This model provides an algorithmic level description of the task, which in turn provides numerical values that could conceivably be identified in neural signals in future recording studies. It's one thing to show that caching birds have flexible memory (as past work has shown), but what seems uniquely new about this paper is that it leverages this past work towards a model that makes instant by instant predictions for how 'special' places influence behavior in a space- and time-dependent way. This algorithmic level description of the behavior is what's novel, which should therefore be emphasized in the abstract/intro.

Discussion paragraph lines 375-387 provides the most accurate description of the contribution of the present work.*Reviewer #2 (Recommendations for the authors):*

In this article the authors present a new behavioural task in birds and combine it with a modelling approach to show that birds can use food-caching memories to guide both the retrieval of cached food as well as to guide the decision where to cache new food. Overall this is an interesting study and with mainly only minor issues.

The authors claim in the abstract that they could show that a single memory can be used for at least two unrelated goals. However, in this case the goals are not really unrelated, in my view. Caching new food and retrieving old food is still part of one behavioural aspect/goal of the animal.

Currently the authors separately present first that birds have individual spatial biases and later on with a modelling approach show that previous visits and general proximity also biases the animals search and caching behaviour. But how do the two aspects relate to each other? One would assume that a general individual bias would decrease the number of possible locations the animal in general has to look at and therefore it is perhaps a mix of memories being used: the bias memory as well as the episodic memory of caching food. If one would include the bias as well as the principles that the animals have a proximity preference, how much variance will still be explained by the episodic memory? Or is the individual bias a result of the proximity bias? It would be good if the authors could create one major model which includes all identified factors to see how much which factor contributes.

Further, it would help readers to understand the details of the behaviour if for example videos of the bird behaviour are provided with the manuscript for the different types of trials. That would help the reader understand in which times scales the animal is remembering and deciding. Further, an overall schematic of the timeline for the birds would be helpful. How long were they trained and habituated on what? How many sessions per day? How many days/weeks in total? For example days/weeks could go to the right and then down would be time within a day since multiple sessions were done per day. Then it can also be marked which data was included for analysis.

The authors did not provide adequate statistical information related to each reported p-value. Example: Line 117 → "In all birds, entropy was lower than expected by chance p < 0.001", they did not clarify what is the chance probability and which statistical test they used to compute the p-value. I assume they computed it based on the shuffling analysis (lines 629 and 630) but it needs to be clearly and more adequately described.*Reviewer #3 (Recommendations for the authors):*

In this manuscript Applegate and Aronov used statistical modeling to study the food caching strategies of the black-capped chickadee in the laboratory setting. The authors show that in a modest sample of birds a subset prefer the center of their arena (61cm x 61cm). Birds tended to have spatially autocorrelated behaviors and remembered cached locations as evidenced by avoiding locations that contain a seed while caching and seeking baited locations when retrieving. Birds would often retrieve and recache seeds and these were preferred as were the last seeds cached. In short, the authors can replicate important aspects of natural behaviors in their experimental preparation.

The presented findings clearly set the ground work for careful behavioral analyses of future studies linked to a more mechanistic understanding of why the birds behave as they do. In the present form, the statistical modeling is rather descriptive and the insights gained will likely relate to a rather narrow audience interested in laboratory assessments of spatial memory. In addition, it is known that birds can remember seed caches and the major gains of the present manuscript are not revealing a new faculty but rather describing a method by which to study that faculty in the laboratory. Also, the sample size is rather small (N = 7 to 17) and variable across figures and analyses.

---

## [Author Response]

Essential revisions:(1) All reviewers agreed that the paper overstates the novelty of the behavioral findings. Please address issues raised below.A main concern is to what extent this paper truly offers an original behavioral result with respect to the flexibility of memory. It is known that birds can remember seed caches and the major gains of the present manuscript are not revealing a new faculty but rather describing a method by which to study that faculty in the laboratory. Also, the sample size is rather small (N = 7 to 17) and variable across figures and analyses. For example, many readers will be familiar with decades-old studies by Sherry, Clayton, and Shettleworth that showed similar flexible memory use by caching birds (e.g. Raby et al., 2007; Shettleworth, 1990; Clayton et al., 2003). As is, the current introduction simply states that flexible memory is 'a matter of debate' in animals – but those familiar with previous studies of caching would likely disagree. The authors should re-write the introduction and abstract to be as generous as possible to past field work, to specify precisely what was previously known, and what is the truly new behavioral finding (if any) in the present work. For example, it's conceivable that this paper could be framed differently, i.e. "it has long been known that caching birds exhibit flexible memory that enable them to target specific sites differentially for depositing versus retrieving seeds, here we leverage this flexible memory capacity to quantify precisely how remembered cache locations influence foraging on a moment to moment basis." (or something like this…)As in, a potential strength of the paper is the behavioral modeling which identifies the precise candidate considerations (parameters) that influence moment by moment aspects of foraging. This model has the potential to provide an algorithmic level description of the task, which in turn provides numerical values that could conceivably be identified in neural signals in future recording studies (see point 2 below to shore up this potential). It's one thing to show that caching birds have flexible memory (as past work has shown), but what seems uniquely new about this paper is that it leverages this past work towards a model that makes instant by instant predictions for how 'special' places influence behavior in a space- and time-dependent way. This algorithmic level description of the behavior is what could be novel, which should therefore be emphasized in the abstract/intro.

As the Reviewers suggested, we have significantly re-written both our Abstract and Introduction to refocus the framing of the paper on the power of our behavioral modeling approach. We agree that this is an important contribution of the paper.

We left some discussion of memory flexibility in the revised manuscript, now largely relegated to the Discussion section. We try to be very careful not to state that our paper discovers the flexibility of memory and better cite the relevant literature. “Flexibility” is a broad term, and reviewers correctly mention several published papers that show some forms of flexibility, both during caching and during retrieval. For example, birds prefer that contain specific types of food and are able to change this preference in different contexts. Birds can also choose caching locations based on knowledge of the environment.

Our results demonstrate a very specific type of flexibility: the use of the same exact memory to both cache and retrieve food. To our knowledge, this has not been shown in food-caching birds. Although chickadees are known to avoid caching into occupied sites, it has not been known that this behavior depends on memory. We feel that this result is important and will be interesting to scholars of memory because caching and retrieving are very distinct behaviors, and the use of memory during these behaviors serves different purposes. Avoidance of occupied sites during caching is likely a defense against pilfering, whereas attraction to these sites during retrieval serves immediate feeding needs.

We think that this type of flexibility beautifully complements the examples of flexibility described in previous studies of food caching, and we tried to clarify these points carefully in the revised Discussion. We do, however, now focus most of the paper on the contributions of our behavioral modeling.

(2) If, as the reviewers suggest, the focus of the paper shifts from a descriptive account of the behavior in the new preparation to an emphasis on the validity of the statistical models, then more needs to be done to motivate the modeling decisions. It is currently not convincing that the parameters of the model are dissociable, especially since birds can continually cache and retrieve and recache.Here are some unanswered questions that should be addressed if we are to take the quantitative parameter estimates of the models seriously, rather than the overall goodness of fit revealing the existence of some feature of the behavior (e.g. behavior is spatially autocorrelated).(2.1.) Is log-linear interaction of the parameters appropriate or are there more complex dependencies?

We addressed this question in the revised manuscript by fitting additional models (new Models #12-13) to account for interactions between pairs of parameters. In other words, rather than modeling log-probability only as a weighted sum of several factors, we also included second-order terms for the products of two of these factors. We tested each pair of factors separately, and used our standard model comparison to ask whether including an interaction term for those two factors justified the addition of an extra parameter.

We now include an additional supplementary figure (Figure 3—figure supplement 1) that shows all of the new results related to considering interaction terms between parameters. Briefly, these results are:

1) We indeed found one nonlinear interaction between parameters that improved the model fit: birds preferred to cache into sites that were *both* empty *and* close to the previously visited location. The preference for sites that had both of these characteristics was supralinear – i.e., not entirely accounted by the linear model.

2) However, when we accounted for this supralinear interaction and examined "mnemonic" parameters γocc and γemp (new Model #13 and #12 respectively), the best-fit values of these parameters were nearly identical to those in the original linear model (Model #3). Therefore, using a simpler linear model did not distort the main conclusions of the paper regarding the use of memory.

(2.2) Is a Gaussian spatial autocorrelation function appropriate?

We addressed this question by including a new supplementary figure that explores alternative autocorrelation functions (Figure 3—figure supplement 2) and by clarifying why we chose a Gaussian spatial autocorrelation to begin with. Note that the manuscript uses a *notched* Gaussian autocorrelation function: a function that is Gaussian for distance>0, but has a value of 0 at distance=0.

1) In the new supplementary figure, we show the actual autocorrelation functions – i.e., the probabilities of caching and checking at various distances away from the previous site. These plots indeed qualitatively look like notched Gaussians: in both cases probability decays with distance, but has small values near zero. Therefore, a notched Gaussian is qualitatively not an unreasonable function to use in our models.

2) In the same supplementary figure, we test two other functions that are often used for fitting distance-dependent probabilities: a notched exponential decay (new Model #14) and a Mexican hat function (new Model #15). We find that using a notched exponential does not improve the model fit. However, using a Mexican hat improves the fit, for both caches and checks. This is true for two reasons. First, the notch at zero in the autocorrelation function has a small, but nonzero width, and is therefore better described by the negative component of the Mexican hat function than a δ function. Second, the probability of caching or checking a site at zero distance is actually *lower* than expected by chance. The notch is therefore better fit by a slightly negative amplitude rather than a drop to zero.

3) However, in the same supplementary figure we show that using a Mexican hat function instead of a single Gaussian had no effect on the mnemonic parameters γocc and γemp (new Models #16 and #17). Therefore, using a simpler single-Gaussian model did not distort the main memory-related results of the paper. Given this result, we feel that our new observations about the exact shape of the spatial autocorrelation are interesting, but best presented in a supplementary figure. For clarity of the storyline, we left the simpler Gaussian model in the main figures.

Is the standard deviation fixed across space?

We addressed this question by fitting additional models in which standard deviation is not fixed across space. These models are presented in a new supplementary figure (Figure 3—figure supplement 3). Because walls and feeders are salient objects in the arena, we considered the simple possibility that standard deviation varies with distance from these objects. New Model #18 uses separate standard deviation values for different distances from walls, and Model #19 does so for different distances from feeders.

As Figure 3—figure supplement 3B-C shows, both new models improve the fit in the Caching task, and Model #18 improves the fit in the Retrieval task. However, just as above, we found that using these more complex models had no effect on our mnemonic parameters γocc and γemp (new Models #20 through #23). For clarity, we therefore kept the simpler single-σ models in the main figures of the paper and devoted a supplementary figure to exploring more complex models.

Across strategy?

As we reported in the original manuscript, the best-fit σprv in the Caching task was 15.5 ± 0.9 cm and 12.9 ± 1.7 cm in the Retrieval task. There was no statistically significant difference between these two values (unpaired t-test, p = 0.54). We had not emphasized this point in the original manuscript, but now do so. We also now report the difference in γprv values between the two tasks.

(2.3) Is time-dependent forgetting tested/modeled appropriately? There was an odd mismatch between the statement about preference for re-caching and the lack of a forgetting curve. What is the longest the animals will have a benefit from remembering caches? If only during one session, is this then perhaps a weakness in the task since one would expect birds in the wild to cache for longer time periods? Given that birds are rechecking single sites every 2.4 minutes, the authors should provide a more convincing argument how this unstructured task addresses the issue of memory decay.

We have made the following changes in the revised manuscript to address these questions:

(1) We now emphasize that chickadees exhibited a significant memory effect after an hour even in the experiment where they were removed from the arena and could not recheck the sites (Figure 6—figure supplement 1). We therefore think that memory in our experiment lasts substantially longer than the average 2.4 min between site checks.

(2) We performed an additional analysis, in which we quantified the effect of memory at different time delays. At each time point, we included in the analysis only those sites that had not been checked for a certain duration of time. With increasing duration, this analysis looked at progressively smaller subsets of the data. Yet, we still had statistical power to analyze durations up to about half of the behavioral session, and did not find any significant decrease of the memory effect during this time. This analysis supports the conclusion of point #1 above, and is shown in a new figure (Figure 6—figure supplement 2).

3) Although we did not detect a significant memory decay on the timescale of our experiment, we cannot exclude the possibility of forgetting that occurs afterwards. Our experiments only lasted an hour (in the Caching task), and there was no benefit in remembering site contents for periods longer than that. We therefore softened our language about longevity of memory and emphasize that we do not examine memory lasting longer than 1 h.

(4) We now better highlight published work, which suggest that the majority of chickadee caches in the wild are retrieved within minutes to a few hours. Unlike some other food-caching species, chickadees tend to use caching to maintain a stable food supply over relatively short periods of time. We therefore think that our experiments study memory on a timescale that is ethologically relevant.

(3) Please clarify how the spatial biases identified for individual birds affect caching strategy.Currently the authors separately present first that birds have individual spatial biases and later on with a modelling approach show that previous visits and general proximity also biases the animals search and caching behaviour. But how do the two aspects relate to each other? One would assume that a general individual bias would decrease the number of possible locations the animal in general has to look at and therefore it is perhaps a mix of memories being used: the bias memory as well as the episodic memory of caching food. If one would include the bias as well as the principles that the animals have a proximity preference, how much variance will still be explained by the episodic memory? Or is the individual bias a result of the proximity bias? It would be good if the authors could create one major model which includes all identified factors to see how much which factor contributes.

We think that this might be a misunderstanding, since we already include spatial biases when quantifying other behavioral effects. Effects of proximity and of previous visits are modeled as factors that *alter* the probability given by pbias. I.e., positive values of these factors mean that the corresponding probabilities are higher than expected from the bird’s spatial bias, whereas negative factors mean lower probabilities. So the Reviewer is correct in that the bird’s behavior is driven by a mix of episodic memory and biases (which may or may not be memory-guided). We think that this issue can be fixed by adding some clarifying language to the main text, which we did in Lines 178-180.

(4) Please provide supplemental videos and increased clarity of the experimental timeline.Further, it would help readers to understand the details of the behaviour if for example videos of the bird behaviour are provided with the manuscript for the different types of trials. That would help the reader understand in which times scales the animal is remembering and deciding.

We have now included supplemental videos featuring a 1-min fragment from the Caching task and an example trial from the Retrieval task (Supplemental videos 1,2). To give the reader a better sense of the time scales, we have also updated the language describing timing both in the main text, as well as in the Methods section. Note that annotations of the entire videos for all sessions will be included on a data sharing server (Dryad) upon publication.

Further, an overall schematic of the timeline for the birds would be helpful. How long were they trained and habituated on what? How many sessions per day? How many days/weeks in total? For example days/weeks could go to the right and then down would be time within a day since multiple sessions were done per day. Then it can also be marked which data was included for analysis.

More clarification of the experimental timeline is now found in the methods section. We now describe the duration of each habituation step in a more organized format. Briefly, birds were run daily, once per day, for a series of several weeks.

(5) Several technical points, if addressed, could make the statistics a bit stronger.First, for the description of spatial biases, it appears that the same data is used to assign cluster membership and then quantify cluster separation. Some sort of cross-validation should be used throughout the analyses plotted in Figure 2 (Panel 2E notwithstanding).

Instead of using an arbitrary decision boundary, we now perform k-means clustering to group the data in Figure 2 (Lines 712). Additionally, we used cross-validation to show that birds reliably fall into the two clusters. Here, for every bird, we excluded that bird from the data, recomputed k-means clustering, then assigned the left-out bird to the cluster with the nearest centroid. In all 10 birds, this resulted in identical cluster assignments. The cross-validation is shown in a new supplemental figure (Figure 2—figure supplement 1).

Second, for the model fitting, a point estimate is used. It would be good to know the confidence around these point estimate and here a Bayesian framework would be helpful perhaps using Markov chain Monte Carlo methods to finds highest density intervals for likely parameter estimates (e.g. Annis and Palmeri 2017, doi: 10.1002/wcs.1458; Kruschke 2013 DOI: 10.1037/a0029146).

We believe that, for the population, this issue is addressed by showing the value of each parameter separately for each bird (as individual symbols in the figures), in addition to reporting the median and standard error. To provide some measure of confidence for each bird, we now also report the fraction of birds that were individually significant for each of the parameters we tested (Table 2). We measured this confidence by bootstrapping each bird’s individual sessions and refitting the parameters.